# Tubular cell polyploidy protects from lethal acute kidney injury but promotes consequent chronic kidney disease

Letizia De Chiara[1], Carolina Conte[1], Roberto Semeraro[2], Paula Diaz-Bulnes[3], Maria Lucia Angelotti[1], Benedetta Mazzinghi[4], Alice Molli[1,4], Giulia Antonelli[1], Samuela Landini [5], Maria Elena Melica[1], Anna Julie Peired [1], Laura Maggi [2], Marta Donati[1], Gilda La Regina[1], Marco Allinovi[6], Fiammetta Ravaglia [7], Daniele Guasti[8], Daniele Bani [8], Luigi Cirillo [1,4], Francesca Becherucci[4], Francesco Guzzi [7], Alberto Magi[9], Francesco Annunziato [2,10], Laura Lasagni[1], Hans-Joachim Anders[11], Elena Lazzeri [1,12] ✉ & Paola Romagnani [1,4,12] ✉

Acute kidney injury (AKI) is frequent, often fatal and, for lack of specific therapies, can leave survivors with chronic kidney disease (CKD). We characterize the distribution of tubular cells (TC) undergoing polyploidy along AKI by DNA content analysis and single cell RNA-sequencing. Furthermore, we study the functional roles of polyploidization using transgenic models and drug interventions. We identify YAP1-driven TC polyploidization outside the site of injury as a rapid way to sustain residual kidney function early during AKI. This survival mechanism comes at the cost of senescence of polyploid TC promoting interstitial fibrosis and CKD in AKI survivors. However, targeting TC polyploidization after the early AKI phase can prevent AKI-CKD transition without influencing AKI lethality. Senolytic treatment prevents CKD by blocking repeated TC polyploidization cycles. These results revise the current pathophysiological concept of how the kidney responds to acute injury and identify a novel druggable target to improve prognosis in AKI survivors.

Acute kidney injury (AKI) is a syndrome characterized by an acute deterioration of kidney function impacting 20.0-31.7% of hospitalized patients and 8.3% of outpatients[1]. With no specific medical treatment available, AKI is potentially lethal and hence a global health concern[1]. In addition, AKI survivors face a high risk of chronic kidney disease (CKD)[2], referred to as "AKI-CKD transition" that is associated with a high cardiovascular risk and the likelihood of progression toward end stage kidney disease (ESKD)[3–5]. The traditional concept of kidney function recovery after AKI assumes a widespread proliferative capacity and regeneration of injured tubular epithelial cells (TC) based on diffuse positivity for cell cycle markers in the kidney, which is, however, inconsistent with the high incidence of CKD after AKI[6,7]. We recently reported that these cell cycle markers do not faithfully indicate cell proliferation but rather cell cycle entry to initiate polyploidization in TC in response to AKI[8]. Polyploidization is the process that leads a normally diploid cell to acquire additional set(s) of

[1]Department of Experimental and Clinical Biomedical Sciences "Mario Serio", University of Florence, Florence 50139, Italy. [2]Department of Experimental and Clinical Medicine, University of Florence, Florence 50139, Italy. [3]Translational immunology, Instituto de Investigación Sanitaria del Principado de Asturias ISPA, 33011 Oviedo, Asturias, España. [4]Nephrology and Dialysis Unit, Meyer Children's University Hospital, Florence 50139, Italy. [5]Medical Genetics Unit, Meyer Children's University Hospital, Florence 50139, Italy. [6]Nephrology, Dialysis and Transplantation Unit, Careggi University Hospital, Florence 50134, Italy. [7]Nephrology and Dialysis Unit, Santo Stefano Hospital, Prato 59100, Italy. [8]Department of Experimental & Clinical Medicine, Imaging Platform, University of Florence, Florence 50139, Italy. [9]Department of Information Engineering, University of Florence, Florence 50139, Italy. [10]Flow Cytometry Diagnostic Center and Immunotherapy (CDCI), Careggi University Hospital, Florence 50134, Italy. [11]Division of Nephrology, Department of Internal Medicine IV, LMU Hospital, Munich 80336, Germany. [12]These authors contributed equally: Elena Lazzeri, Paola Romagnani. ✉e-mail: elena.lazzeri@unifi.it; paola.romagnani@unifi.it

chromosomes[9]. Most mammalian cells are diploid, but hepatocytes and cardiomyocytes become progressively polyploid during lifetime via an alternative cell cycle named endoreplication[10–13]. In contrast to hepatocytes and cardiomyocytes[10–13], TC remain mostly mononuclear after AKI, explaining why this phenomenon had remained undiscovered[8]. In this study, we aimed to investigate the biological relevance and the structural consequences of TC polyploidization in the kidney, which are currently unknown. We hypothesized that polyploidy in TC is an evolutionary conserved mechanism developed to permit a quick recovery of kidney function after damage assuring survival. However, life-preserving stress responses frequently drive long-term organ failure, as shown in other organs[14–16]. Therefore, we carefully assessed long-term outcomes. To do so, we employed numerous conditional transgenic mouse lines under the control of the Pax8 promoter, which is consistently and exclusively[8,17] expressed by renal TC, to achieve targeted recombination in all the tubular segments.

In this study, we demonstrate that during AKI, YAP1-driven polyploidization of TC acts as an immediate compensatory mechanism to augment residual kidney function and to avoid early death for kidney failure. However, as a trade-off, TC polyploidy promotes TC senescence and progressive interstitial fibrosis, i.e., AKI-CKD transition. Importantly, delayed blockade of YAP1-driven polyploidization can avoid the development of CKD and improve long-term outcome in those that survive the early injury phase. These findings identify a previously unrecognized life-preserving type of cellular response, a Janus-faced role of cell polyploidization in AKI and a novel drug target to improve long-term prognosis in survivors of the acute phase.

## Results

### After AKI, most TC enter the cell cycle and either die or become polyploid

To identify ploidy in mononuclear TC, we combined measurement of the DNA content with assessment of cell cycle phase in Pax8/FUCCI2aR mice by flow cytometry[8] (Fig. 1a, b and Supplementary Fig. 1a–e). Ischemic AKI induced a strong but short-lasting entry into the cell cycle in $39.8 \pm 12.5\%$ of TC at day 2 that became mVenus+ or mCherry+mVenus+ (Fig. 1a, b). At this stage, it was impossible to establish how many of these cycling TC would complete division, endoreplication or eventually die. At day 3 after AKI, the percentage of cells in the G1 phase (i.e., mCherry+ with a 2C DNA content, defined as non-cycling TC) had remained stable in comparison to day 2, whereas the percentage of cycling cells (i.e., mCherry+mVenus+ with DNA content = 2C and mVenus+ cells up to 4C DNA content) had decreased from $39.8 \pm 12.5\%$ to $8.8 \pm 4.7\%$. This suggested that those TC that enter the S phase of the cell cycle at day 2, at day 3 either: (1) keep cycling (2) die or (3) become polyploid via endoreplication. Indeed, considering this distribution as 100%, we found that at day 3, $17 \pm 2.4\%$ of TC (i.e., $38.8 \pm 9.2\%$ of the cells that were cycling at day 2) had acquired ≥4C DNA content (i.e., had become mCherry+, mCherry+mVenus+ with a 4C DNA content or mVenus+ with a DNA content ≥8C), suggesting one or even multiple endoreplication cycles (Fig. 1a–c). By day 5, the percentage of cycling cells was reduced to $1.1 \pm 0.6\%$ remaining consistently low till day 30 ($10.7 \pm 8.7\%$) (Fig. 1b). Polyploid TC decreased from $17 \pm 2.4\%$ at day 3 to $10.9 \pm 1.6\%$ of total TC at day 5, remaining stable thereafter and persisting until day 30 ($12.6 \pm 2.3\%$) (Fig. 1a, b). Of note, virtually all dying cells at the different time points were cycling cells (i.e., mVenus+ or mCherry+mVenus+) based on the FUCCI2aR reporter, suggesting that TC death occurred during the S or G2/M phase of the cell cycle (Fig. 1d). Consistently, $19.6 \pm 13.7\%$ of TC (i.e., $44.4 \pm 15.8\%$ of the TC that were cycling at day 2), died between day 2 and 3 (Fig. 1a, b) and were detectable as dead TC in S or G2/M phase by flow cytometry and confocal microscopy in kidney tissue as well as lost TC in the urine

(Fig. 1d–h). Taken together these results show that, in the early phase of AKI, most TC that start DNA synthesis die and are lost into the urine, while most surviving TC undergo polyploidization within the remaining tubules.

### TC polyploidization occurs distant from the site of injury

The FUCCI2aR reporter is a ubiquitin-based system that permits detection of the cell cycle state of polyploid TC at a specific moment but neither their permanent tracing, underestimating the phenomenon, nor their localization. Indeed, ischemic necrosis affects specifically S3 segments of the proximal tubules in the outer medulla raising the question why AKI involves widespread cell cycle marker positivity all over the cortex (Supplementary Fig. 2a, b)[18,19]. To effectively localize polyploid TC to different tubular segments, we induced AKI in heterozygous Pax8/Confetti mice. As shown in the Supplementary Fig. 2c–g, in these mice the diploid cells recombine only one fluorochrome appearing as mono-coloured cells, while polyploid cells with 4C DNA content recombine two fluorochromes, appearing as bi-coloured or mono-coloured cells[20]. Indeed, when analysing the three fluorochromes that have the same probability of recombination (RFP, CFP and YFP), we can calculate that 2/3 of polyploid cells will recombine two different colours appearing bi-coloured and 1/3 will recombine the same colour twice, appearing mono-coloured, resulting undistinguishable from diploid cells (see Supplementary Fig. 2c–g). Bi-coloured polyploid TC localized mostly in the cortex ($13.3 \pm 2.5\%$ over total coloured TC in the cortex $vs$ $2.7 \pm 0.4\%$ in the outer stripe of outer medulla, Fig. 1i–m and Supplementary Fig. 2h–j), representing a 2.2-fold increase upon AKI in comparison to sham control ($13.3 \pm 2.5\%$ $vs$ $5.9 \pm 0.7\%$). Bi-coloured polyploid TC in the cortex were not only mononucleated but also sporadically binucleated, as confirmed by electron microscopy (Fig. 1n).

To better dissect polyploid TC distribution in the various portions of the nephron, we then performed immunofluorescence staining for: 1. Aquaporin-1 (AQP1) to identify the various segments of the proximal tubule (namely S1, S2 and S3) and the thin descending limb of the Henle's loop; 2. Tamm Horsfall protein (THP) to identify the thick ascending limb of the Henle's loop and the distal straight tubule; 3. Aquaporin-2 (AQP2) to identify the collecting ducts (Fig. 1l–o, Supplementary Fig. 2k–m). S1, S2 and S3 segments were distinguished based on their morphology and localization in distinct kidney areas[21]. Bi-coloured polyploid TC mostly localized to S1 and S2 segments of the cortex sparing the injured outer medulla (Fig. 1l–o, Supplementary Fig. 2k–m). Collectively, diffuse TC polyploidization distant from the injury site suggests a functional role beyond structural replacement of TC lost in the injured outer medulla.

### Polyploid TC arise in a YAP1-related manner

To analyse TC polyploidization, we performed single cell RNA-sequencing (scRNA-seq) on primary cultures of human proximal tubular epithelial cells (hPTC), which we have found to contain a fraction of polyploid hPTC (Supplementary Fig. 3a–d). Unsupervised clustering revealed eleven clusters with distinct expression patterns (Supplementary Fig. 3e, f and Supplementary Table 1). Five clusters were enriched in genes previously associated with different phases of endoreplication-mediated polyploidy: 1. Ribosome biogenesis (cluster 10)[22–24]; 2. Hypertrophy, DNA synthesis and chromatin decondensation (clusters 4, 5, 7)[25,26]; 3. Hippo pathway effector YAP1 targets (cluster 9)[9,27,28] (Supplementary Table 1). In particular, clusters 10, 5, 4, 7 displayed a pattern reported in other organs as characteristic of early, and cluster 9 of late, hypertrophy phases (Fig. 2a)[25,26]. A trajectory analysis along pseudotime suggested that hPTC polyploidization started with increased RNA synthesis and ribosome biogenesis, followed by entry and progression through the cell cycle (Fig. 2b, c). Subsequently, the cells transcribed YAP1 mRNA followed by expression of the

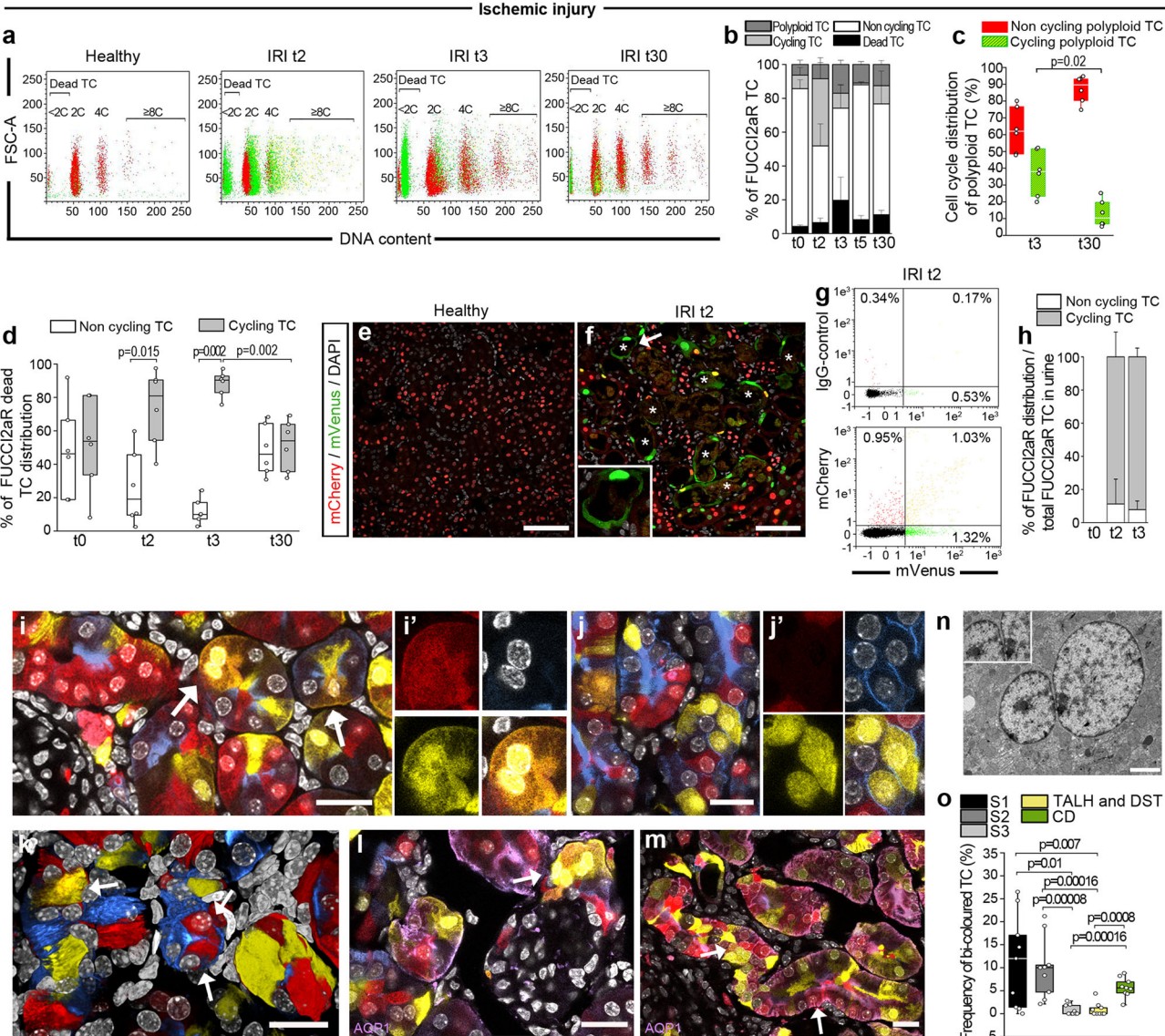

**Fig. 1 | The majority of tubular cells (TC) entering cell cycle becomes polyploid or dies following AKI. a** FACS plots of TC in Pax8/FUCCI2aR mice (*n* = 6) after IRI, showing dead (<2C), diploid (2C), tetraploid (4C) and octaploid or greater (≥8C) TC. Colours match those of the FUCCI2aR reporter. **b** FUCCI2aR TC distribution after IRI. Dead TC: t0*vs*t3 *p* = 0.015, t0*vs*t30 *p* = 0.008; t0*vs*t5 *p* = 0.015, cycling TC: t0*vs*t2 *p* = 0.002, t2*vs*t3 *p* = 0.002, t2*vs*t5 *p* = 0.002, t2*vs*t30 *p* = 0.015; Polyploid TC: t0*vs*t3 *p* = 0.002, t0*vs*t5 *p* = 0.026, t0*vs*t30 *p* = 0.008, t2*vs*t3 *p* = 0.002, t3*vs*t5 *p* = 0.004, t3*vs*t30 *p* = 0.026 (*n* = 6 each time point). **c** Distribution of polyploid TC across cell cycle after IRI (*n* = 6). **d** Distribution of FUCCI2aR dead TC analysed by FACS after IRI (*n* = 6). **e** Representative pictures of **e** healthy and **f** two days after IRI Pax8/FUCCI2aR mice showing necrotic tubu les (*) with mVenus⁺ TC. DAPI counterstains nuclei. Bars 75 μm. The inlet is a higher magnification of the tubule indicated by the arrow. **g** FACS analysis of cells from the urine of Pax8/FUCCI2aR mice after IRI. **h** Distribution across cell cycle of FUCCI2aR TC from the urine of healthy and after IRI mice (multiple urine samples pulled together, *n* = 6 mice for each time

point). **i, j** Heterozygous Pax8/Confetti mice 30 days after IRI. Bars 20 μm. **k** 3D reconstruction of cortical tubules. Bar 20 μm. Arrows indicate bi-coloured TC. AQP1 staining in heterozygous Pax8/Confetti mice 30 days after IRI, arrows show bi-coloured TC in a representative **l** S1 segment and **m** S2 segment. Bars 20 μm. **n** Transmission electron microscopy of a binucleated polyploid TC 30 days after IRI. Bar 2 μm. **o** Bi-coloured TC distribution in tubular segments of heterozygous Pax8/Confetti mice after IRI (*n* = 9). t0: healthy, t2: day 2 after IRI, t3: day 3 after IRI, t5: day 5 after IRI, t30: day 30 after IRI. IRI ischemia reperfusion injury. S1: S1 segment of proximal tubule, S2: S2 segment of proximal tubule, S3: S3 segment of proximal tubule, DST distal straight tubule, TALH thick ascending limb of loop of Henle, CD collecting duct. AQP1 Aquaporin-1. Statistical significance was calculated by two-sided Mann-Whitney test; numbers on graphs represent exact *p* values. Data are expressed as mean ± SEM in graph 1b and 1h. Box-and-whisker plots: line = median, box = 25–75%, whiskers = outlier (coef. 1.5).

endoreplication-specific regulators E2F1, E2F7 and E2F8[9] and YAP1 target genes, particularly CTGF[28,29] (Fig. 2c), while TAZ mRNA expression was not altered (Supplementary Fig. 3g). Sorting of mCherry-hPTC (i.e., hPTC transduced with a mCherry-G1 vector) with ≥4C DNA content (polyploid hPTC) showed enrichment for the YAP1 downstream target CTGF and the SEMA5A[30] gene, characteristic of hypertrophic cluster 9 (Fig. 2d–f and Supplementary Fig. 3h–j). Blocking YAP1 nuclear translocation with verteporfin[31] significantly decreased the

percentage of polyploid mCherry-hPTC further suggesting that YAP1 controls hPTC polyploidy in vitro (Fig. 2g, h and Supplementary Fig. 3k). However, as verteporfin blocks nuclear translocation also of the YAP1 paralog TAZ to some extent (Supplementary Fig. 3l), we silenced YAP1 and TAZ independently in mCherry-hPTC. These experiments confirmed that only YAP1 significantly reduced mCherry-hPTC polyploidy (Supplementary Fig. 3m–q). Next, we performed scRNA-seq analysis on healthy mouse kidney and on kidneys at days 2

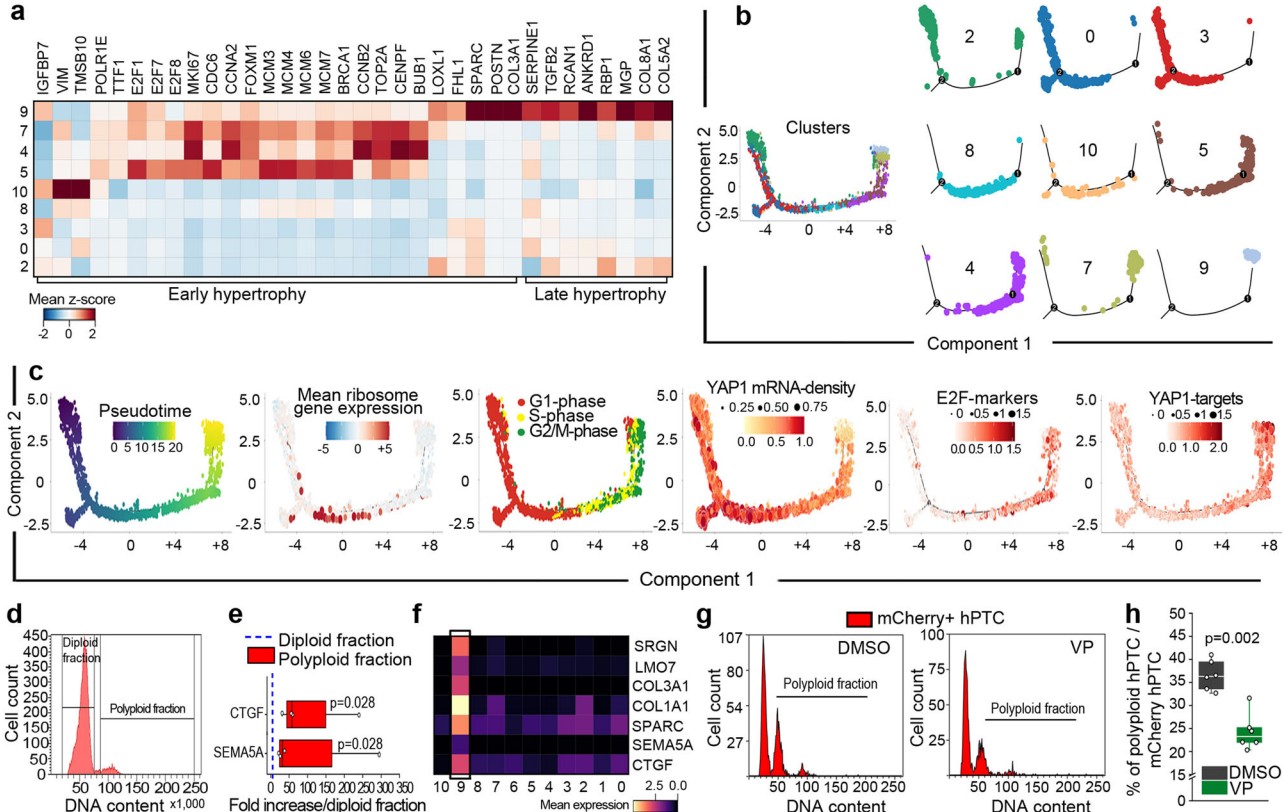

**Fig. 2 | Polyploidization of human proximal tubular cells (hPTC) is controlled by YAP1. a** Matrix plot of hPTC clusters showing hypertrophy genes. **b** Monocle2 pseudotime trajectory of hPTC clusters and "facet" trajectory (right) showing cluster distribution in order of appearance. **c** Monocle2 trajectory of hPTC coloured by pseudotime, mean ribosome gene expression difference, cell cycle phases, YAP1 mRNA accumulation, E2F-markers (E2F1, E2F7, E2F8) and YAP1-targets (ANKDR1, BIRC5, CTGF, CDK1, CCNB1, AKT1). **d** Cell cycle distribution of mCherry-

hPTC. **e** Gene expression of sorted polyploid mCherry-hPTC over diploid mCherry-hPTC (n = 4). **f** Matrix plot of cluster 9 characteristic genes. **g** Cell cycle distribution of mCherry-hPTC treated with DMSO or with verteporfin (VP). A representative experiment out of 6 is shown. **h** Percentage of polyploid mCherry-hPTC in DMSO-treated or VP-treated culture (n = 6). Statistical significance was calculated by two-sided Mann-Whitney test; numbers on graphs represent exact p values. Box-and-whisker plots: line = median, box = 25–75%, whiskers = outlier (coef. 1.5).

and 30 after ischemic injury (Supplementary Fig. 4a, b). We identified and further analysed proximal (P) TC on day 2 and 30 after injury, when polyploidization occurs (Fig. 3a, b, Supplementary Fig. 4c–e and Supplementary Table 2). In vivo, scRNA-seq analysis on PTC at day 2 after AKI revealed high levels of ribosome biogenesis and enrichment in hypertrophy genes, culminating in cluster 8 (Fig. 3c, d). More importantly, in vivo cluster 8 closely resembled the signature already observed in polyploid hPTC in vitro (Supplementary Tables 1, 2, highlighted genes) and was enriched with endoreplication-specific regulators E2f1, E2f7 and E2f8 and YAP1 target genes[9,28,29] (Fig. 3e–g). A trajectory analysis along pseudotime showed similar results to those obtained in vitro with increased ribosome biogenesis, followed by entry and progression through the cell cycle, culminating in the expression of endoreplication-specific regulators E2f1, E2f7, E2f8 and YAP1 target genes (Fig. 3h, i). The closely related cluster 9 from kidneys at day 30 after AKI was also enriched in hypertrophy genes (Fig. 3g). Interestingly, increased ribosomal synthesis was accompanied by a progressive increase of transcriptome abundance along the trajectory in endoreplicating clusters, confirming that polyploid cells are hypertrophic cells (Fig. 3j, k). Protein analysis showed that nuclear translocation of YAP1 started immediately after AKI and persisted until day 30 (Supplementary Fig. 5a, b). Conversely, TAZ expression was not significantly altered following AKI (Supplementary Fig. 5c, d). These results demonstrate that a subset of PTC after AKI shows polyploidization-related hypertrophy signature and that PTC undergo polyploidization in a YAP1-related manner.

## YAP1 controls TC polyploidization through E2F7, E2F8 and AKT1
We next tried to establish the mechanism through which YAP1 triggers polyploidy in TC. Firstly, we screened the genes previously shown to be involved in endoreplication[9,27] and enriched in the polyploid clusters that emerged from the scRNA-seq analysis in vitro and in vivo. Upon verteporfin treatment, E2F7, E2F8 and AKT1 were shown to be the most strongly downregulated in mCherry-hPTC (Fig. 4a), suggesting they may be directly regulated by YAP1 activation. As verteporfin partially affects TAZ localization along with YAP1, we then confirmed that only YAP1 knock-down and not TAZ knock-down successfully downregulated E2F7, E2F8 and AKT1 in mCherry-hPTC (Fig. 4b). Importantly, chromatin immunoprecipitation (ChIP) assay proved that YAP1 directly binds to the promoters and activates all of them, with E2F8 showing the strongest enrichment, as well as CTGF which was used as an internal control[29] (Fig. 4c–f). Accordingly, silencing of E2F7, E2F8 and AKT1 resulted in a significant downregulation of polyploidy in mCherry-hPTC, replicating the results obtained with YAP1 silencing (Fig. 4g–i). Collectively, these results demonstrate that YAP1 controls polyploidization of TC via E2F7, E2F8 and AKT1.

## TC polyploidization preserves kidney function assuring survival after AKI
To determine the functional role of YAP1-dependent polyploidization, we used mice with a TC-specific YAP1 depletion upon exposure to doxycycline (Pax8/YAP1*ko*, Supplementary Fig. 6a–h). YAP1 was successfully knocked-out in TC upon doxycycline-induced recombination

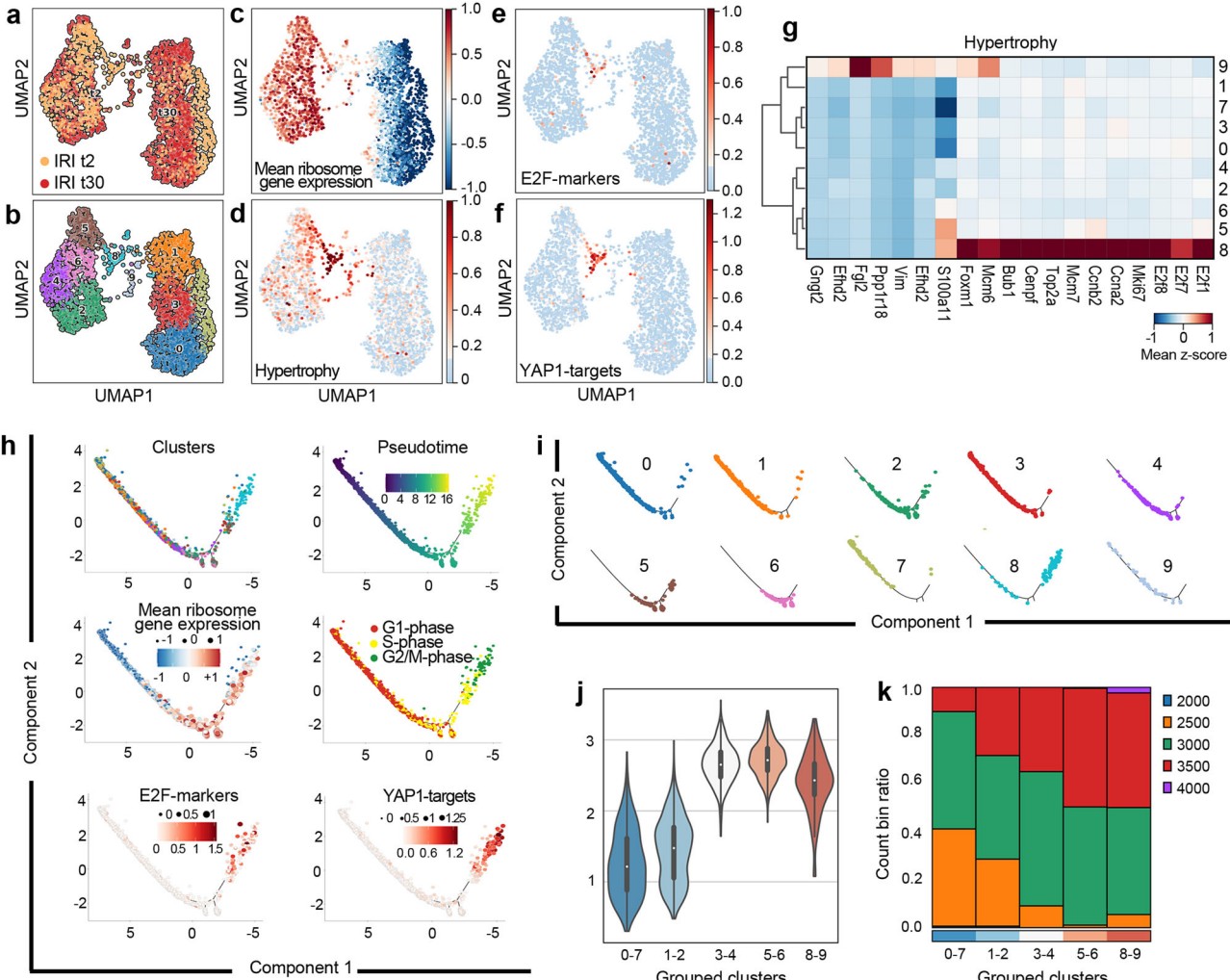

**Fig. 3 | Polyploidization of proximal tubular cells (PTC) after AKI in mice is controlled by YAP1. a** UMAP of experimental time distribution and **b** cluster distribution of mouse PTC at day 2 (t2) and 30 (t30) after IRI. **c** UMAP showing ribosome gene expression difference and **d** hypertrophy genes (Top2a, Mcm7, Ccna2, Ccnb2, Tmsb10, S100a11). **e** UMAP showing E2F-markers (E2f1, E2f7, E2f8) and **f** YAP1-targets (Ankdr1, Birc5, Ctgf, Cdk1, Ccnb1, Akt1) in PTC. **g** Matrix plot of PTC at 2 and 30 days after IRI, showing hypertrophy genes. **h** Monocle2 trajectory of PTC clusters and trajectory of PTC coloured by pseudotime, mean ribosome gene expression difference, cell cycle phases, E2F-markers (E2f1, E2f7, E2f8) and

YAP1-targets (Ankdr1, Birc5, Ctgf, Cdk1, Ccnb1, Akt1). **i** Monocle2 "facet" trajectory showing cluster distribution in order of appearance. **j** Violin plots showing ribosome distribution in clusters grouped by ribosome gene expression. Median distribution is represented by the white dot; the black bar in the centre of the violin represents the interquartile range between the first and third quartile; the black lines stretched from the bar represent the lower/upper adjacent values defined as first interquartile −1.5 and third interquartile +1.5, respectively. **k** Bar plot showing the binned normalized counts distribution in each cluster group. IRI ischemia reperfusion injury.

(Supplementary Fig. 6i, j) while TAZ expression was not affected (Supplementary Fig. 6k, l). These mice developed less cycling TC two days after unilateral ischemia reperfusion injury (ischemic injury) and less polyploid TC at day 3 in comparison to Pax8/WT mice (Fig. 5a–c) and showed a more pronounced decline of kidney function at day 2 (Fig. 5d). However, the unilateral ischemic model cannot truly evaluate the role of polyploidization on kidney function recovery due to the confounding effect of the contralateral kidney during compensation. To address this point, we used a model of nephrotoxic AKI with bilateral kidney involvement. Pax8/YAP1*ko* mice showed a lower percentage of cycling TC and of polyploid TC after AKI (Fig. 5e–g), while TC death and number were comparable (Fig. 5h, i). We confirmed that only YAP1 upregulation and not TAZ was impaired in Pax8/YAP1*ko* mice 2 days after nephrotoxic injury in comparison to Pax8/WT mice (Supplementary Fig. 6m–p). Importantly, this model identified YAP1-driven TC polyploidy as a survival mechanism protecting from hyperkalemia and uremic death (Fig. 5j–l). Indeed, at 48 h, the injured kidneys of Pax8/YAP1*ko* mice, as well as the TC, were smaller compared

to WT mice (Fig. 5m–o), consistent with the concept that hypertrophy of cortical TC enlarges kidney dimensions during AKI, a phenomenon well-known clinically. Considering that Pax8 is selectively expressed in TC and that TC death and number were comparable between WT and KO mice, the reduced survival of Pax8/YAP1*ko* mice can only be related to a defect in polyploidization-mediated hypertrophy of TC. Altogether, these data show that YAP1-driven polyploidization-dependent hypertrophy of cortical TC is a previously unknown protecting mechanism of immediate cellular adaptation that sustains residual kidney function during the early and potentially fatal phase of AKI.

**YAP1-driven TC polyploidization leads to CKD**
Next, we studied the consequences of sustained TC polyploidization in a conditional transgenic mouse model of continuous YAP1 activation, referred to as Pax8/SAV1*ko* mice (Supplementary Fig. 7a–e)[32]. Persistent YAP1 activation in TC increased kidney size (Fig. 6a–c) without affecting the absolute number of FUCCI2aR TC (Supplementary Fig. 7f) whereas TAZ was not upregulated (Supplementary

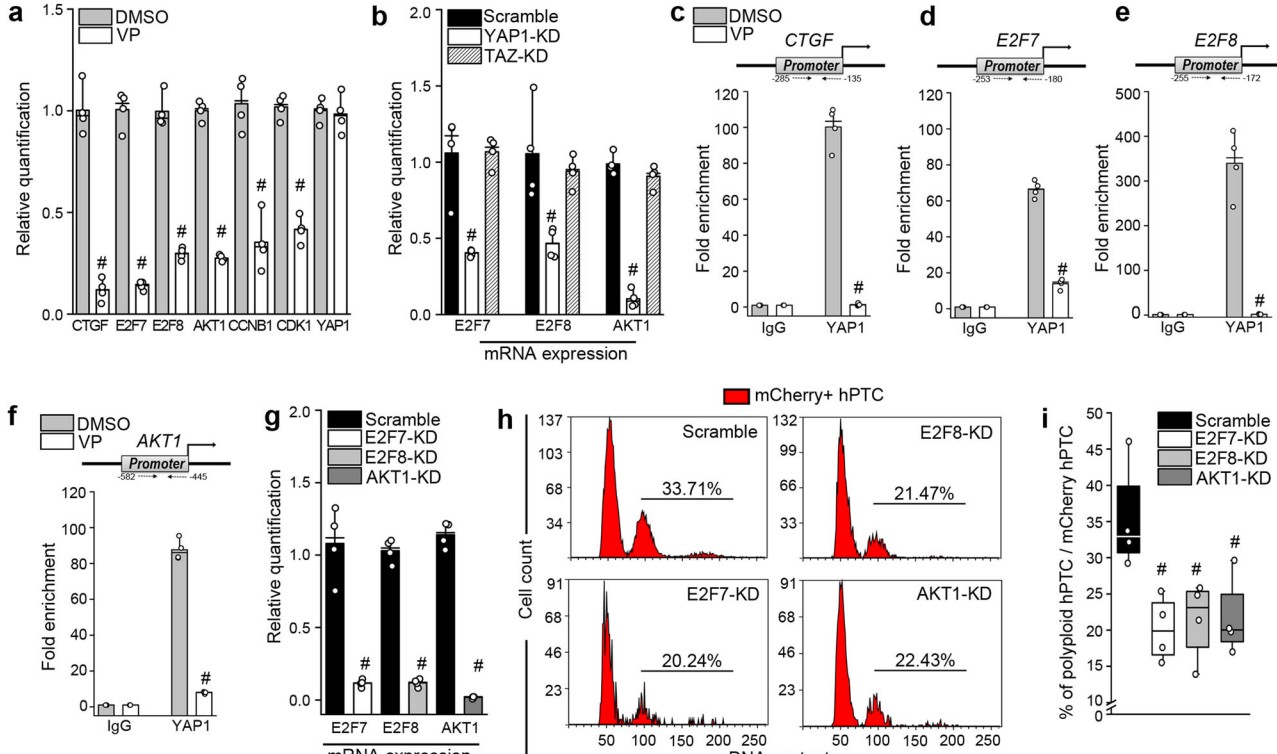

**Fig. 4 | YAP1 controls polyploidization via AKT1, E2F7 and E2F8 activation.**
**a** qRT-PCR analysis of YAP1 targets in DMSO and verteporfin (VP) treated mCherry-hPTC ($n = 4$). CTGF was used as a positive control. YAP1 was used as a negative control. # Significance between DMSO and VP treated mCherry-hPTC for each gene analysed, $p = 0.028$. **b** qRT-PCR quantification of E2F7, E2F8 and AKT1 in scramble, YAP1, and TAZ knock-down (KD) mCherry-hPTC ($n = 4$). # Significance between scramble and YAP1-KD mCherry-hPTC for each gene analysed, $p = 0.028$. Scramble and TAZ-KD conditions are not significantly different. Chromatin immunoprecipitation assay showing YAP1 binding on **c** CTGF, **d** E2F7, **e** E2F8 and **f** AKT1 promoters in DMSO and VP treated mCherry-hPTC ($n = 4$). #Significance between DMSO and VP treated mCherry-hPTC, $p = 0.028$. **g** KD efficiency of E2F7, E2F8 and AKT1 GapmeRs ($n = 4$). #Significance between scramble and KD mCherry-hPTC, $p = 0.028$. **h** Cell cycle distribution of mCherry-hPTC transfected with scramble, E2F7, E2F8 and AKT1 GapmeRs. A representative experiment out of 4 is shown. **i** Percentage of polyploid mCherry-hPTC in scramble-treated, E2F7-KD, E2F8-KD and AKT1-KD cultures ($n = 4$). # Significance between scramble and KD mCherry-hPTC, $p = 0.028$. hPTC: human proximal tubular cells. Statistical significance was calculated by two-sided Mann-Whitney test. Box-and-whisker plots: line = median, box = 25–75%, whiskers = outlier (coef. 1.5). Bar plots: line = mean, whisker = outlier (coef. 1.5).

Fig. 7g, h). By contrast, persistent YAP1 activation in TC increased the percentage of polyploid TC already in healthy conditions (Fig. 6d–f). This was associated with prominent interstitial fibrosis compared to WT controls (Fig. 6g–j). Enhanced TC polyploidy in Pax8/SAV1*ko* mice associated with expression of β-galactosidase, a marker of cell senescence (Fig. 6k, l). All these changes were accompanied by a progressive glomerular filtration rate (GFR) decline (Fig. 6m). Thus, induced overexpression of YAP1 in TC of healthy mice drives polyploidization, interstitial fibrosis and senescence of TC, i.e., progressive CKD.

## TC polyploidization aggravates AKI-CKD transition but attenuates AKI

We then evaluated how enhanced polyploidization of TC in Pax8/SAV1*ko* mice affects the different phases of AKI. At day 30 after unilateral ischemic injury we did not observe any difference in TC number in Pax8/SAV1*ko* compared to kidneys of Pax8/WT mice (Supplementary Fig. 8a). However, polyploidization, interstitial fibrosis and TC senescence were enhanced in ischemic Pax8/SAV1*ko* mice in comparison to ischemic Pax8/WT mice, indicating that YAP1-activated polyploidization promotes AKI-CKD transition (Supplementary Fig. 8b–j). Of note, in ischemic Pax8/SAV1*ko* mice polyploidization, interstitial fibrosis and TC senescence were enhanced also in comparison to healthy Pax8/SAV1*ko* mice, indicating that also AKI significantly contributes to CKD (Supplementary Fig. 8b–j). In addition, we observed binucleated TC at electron and confocal microscopy, as already

observed in Pax8/Confetti mice with normal YAP1 function (Fig. 7a and Supplementary Fig. 8c, d). Polyploid TC kept cycling and produced TC with 8C or more of DNA content at 30 days prominently in Pax8/SAV1*ko* mice (Fig. 7b, c). Re-analysis of scRNA-seq data showed that polyploid TC cluster(s) displayed a profibrotic and senescent signature, both in vitro and in vivo (Fig. 7d and Supplementary Figs. 8k, 9a–f). In addition, we observed enhanced TC polyploidization, interstitial fibrosis, and TC senescence in Pax8/SAV1*ko* mice also in the nephrotoxic AKI (Fig. 7e–k and Supplementary Fig. 8l). Consistently, Pax8/SAV1*ko* mice showed an attenuated loss of kidney function at day 2 after AKI but developed a more severe CKD post-AKI, in comparison to both Pax8/WT mice that underwent nephrotoxic AKI and healthy Pax8/SAV1*ko* mice as well as an increased survival rate (Fig. 7l, m). Thus, enhanced TC polyploidization sustains residual kidney function and improves survival during early AKI but subsequently promotes AKI to CKD transition.

## TC polyploidy in human biopsies correlates with kidney fibrosis after AKI

To validate these findings in humans, we investigated the presence of TC polyploidization in 45 patients who underwent diagnostic kidney biopsy after one or more AKI episodes (Supplementary Table 3) and categorized them into "early" and "late" according to the AKI-to-biopsy interval (see Methods). Co-expression of nuclear p-H3 and perinuclear CDK4 identified cells in G1 with chromatin remodeling, i.e., polyploid cells (Fig. 8a, b)[8,33]. Their quantification in the tubule (stained with

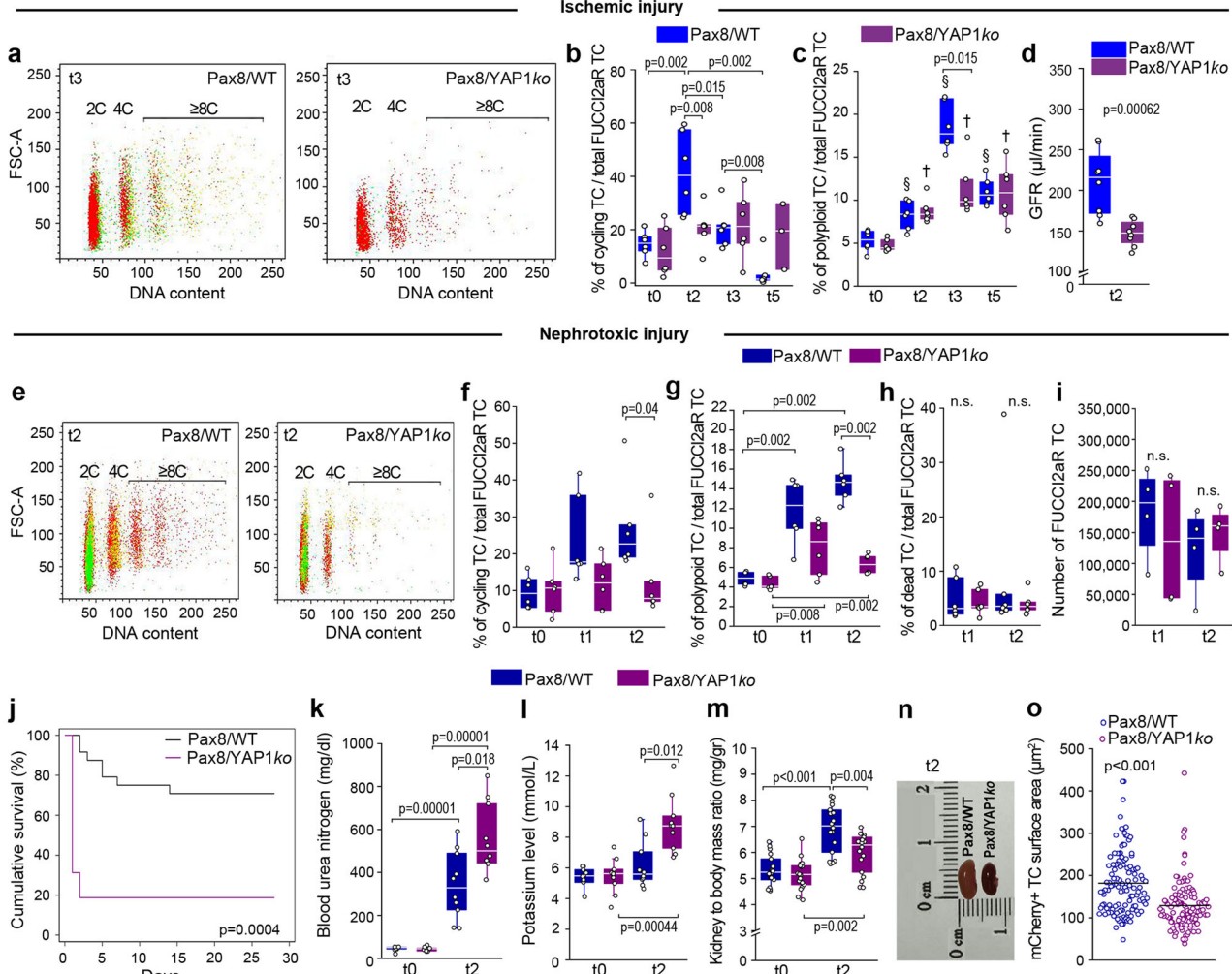

**Fig. 5 | Early tubular cell (TC) polyploidization preserves residual kidney function and assures survival during AKI. a** FACS plots of Pax8/FUCCI2aR (Pax8/WT) (*n* = 6) and Pax8/FUCCI2aR/YAP1ko (Pax8/YAP1*ko*) (*n* = 6) showing diploid (2C), tetraploid (4C) and octaploid or greater (≥8C) TC. Colours match the FUCCI2aR reporter. **b** Percentage of cycling TC in healthy (t0) mice and after IRI (*n* = 6). **c** Percentage of polyploid TC in healthy (t0) mice and after IRI (*n* = 6). (§) Significance within Pax8/WT mice: t0*vs*t2 *p* = 0.015, t0*vs*t3 *p* = 0.002, t0*vs*t5 *p* = 0.002, t2*vs*t3 *p* = 0.002, t2*vs*t5 *p* = 0.026, t3*vs*t5 *p* = 0.002. (†) Significance within Pax8/YAP1*ko* mice: t0*vs*t2 *p* = 0.002, t0*vs*t3 *p* = 0.002, t2*vs*t3 *p* = 0.04. **d** Glomerular filtration rate (GFR) measurement (*n* = 8). (t0 = healthy, t2 = day 2 after IRI, t3 = day 3 after IRI, t5 = day 5 after IRI). **e** FACS plots of Pax8/WT (*n* = 6) and Pax8/YAP1*ko* (*n* = 6) TC after nephrotoxic injury, showing diploid (2C), tetraploid (4C) and octaploid or greater (≥8C) TC. **f** Percentage of cycling TC in healthy (t0) mice and after nephrotoxic injury (*n* = 6). **g** Percentage of polyploid TC in healthy mice (t0)

and after nephrotoxic injury (*n* = 6). **h** Percentage of dead TC after nephrotoxic injury (*n* = 6). **i** Total FUCCI2aR TC number after nephrotoxic injury (*n* = 4). **j** Survival analysis of mice after nephrotoxic injury. Kaplan-Meier analysis showed a significant difference at Log rank comparison X2 = 17.663, *p* = 0.0004 (*n* = 24 Pax8/WT, *n* = 14 Pax8/YAP1*ko*, none censored). **k** Blood urea nitrogen measurement (*n* = 10). **l** Potassium level in the serum (*n* = 10). **m** Kidney weight (*n* = 18), *p* = 0.0000016. **n** Picture of Pax8/WT and Pax8/YAP1*ko* kidneys 2 days after nephrotoxic injury. **o** Cell surface area of mCherry+ TC at day 1 and 2 after nephrotoxic injury. 30 TC for each mouse (t1 *n* = 2; t2 *n* = 2) were counted. (t0 = healthy, t1 = day 1 after nephrotoxic AKI, t2 = day 2 after nephrotoxic AKI, *p* = 5.4 × 10⁻¹⁰). Statistical significance was calculated by two-sided Mann-Whitney test; numbers on graphs represent exact *p* values or are provided in the legend. Box-and-whisker plots: line = median, box = 25–75%, whiskers = outlier (coef. 1.5).

phalloidin) showed a substantial increase of polyploid TC in kidney biopsies of AKI patients in comparison to healthy kidneys (Fig. 8c). The percentage of polyploid TC in biopsies obtained early after AKI was significantly higher in comparison to the percentage in biopsies taken late after AKI (Fig. 8d), in line with the decrease of polyploidization observed overtime after AKI in mice. In addition, in early biopsies, DNA content was significantly increased in YAP1⁺ nuclei in comparison to YAP1⁻ nuclei indicating that where YAP1 is active TC have an increased DNA content (i.e., they are polyploid) (Fig. 8e, f). Importantly, level of polyploid TC correlated with fibronectin expression in kidney biopsies of patients of the "late" group (Fig. 8g–i). Taken together, these results demonstrate that the extent of TC polyploidization associates with kidney fibrosis after AKI also in humans.

## Delayed onset of YAP1-inhibition attenuates AKI-CKD transition

We then speculated that delayed YAP1 inhibition could prevent AKI-CKD transition without affecting AKI survival. We employed the selective inhibitor of YAP1/TEAD transcriptional activity CA3[34,35] and started treatment of Pax8/FUCCI2aR and non-induced (WT) mice 4 days after nephrotoxic AKI, i.e., after recovery of function loss had occurred (Fig. 9a). Delayed treatment with CA3 prevented AKI-CKD transition, as defined by kidney function at day 30 (Fig. 9a, b). Percentages of non-cycling polyploid TC (i.e., polyploid TC in G1) compared to vehicle controls were similar at day 30 (Fig. 9c), suggesting that they had been generated before the start of CA3 treatment and were growth-arrested. By contrast, cycling polyploid TC were reduced, proving that inhibiting YAP1 blocked further endoreplication cycles of

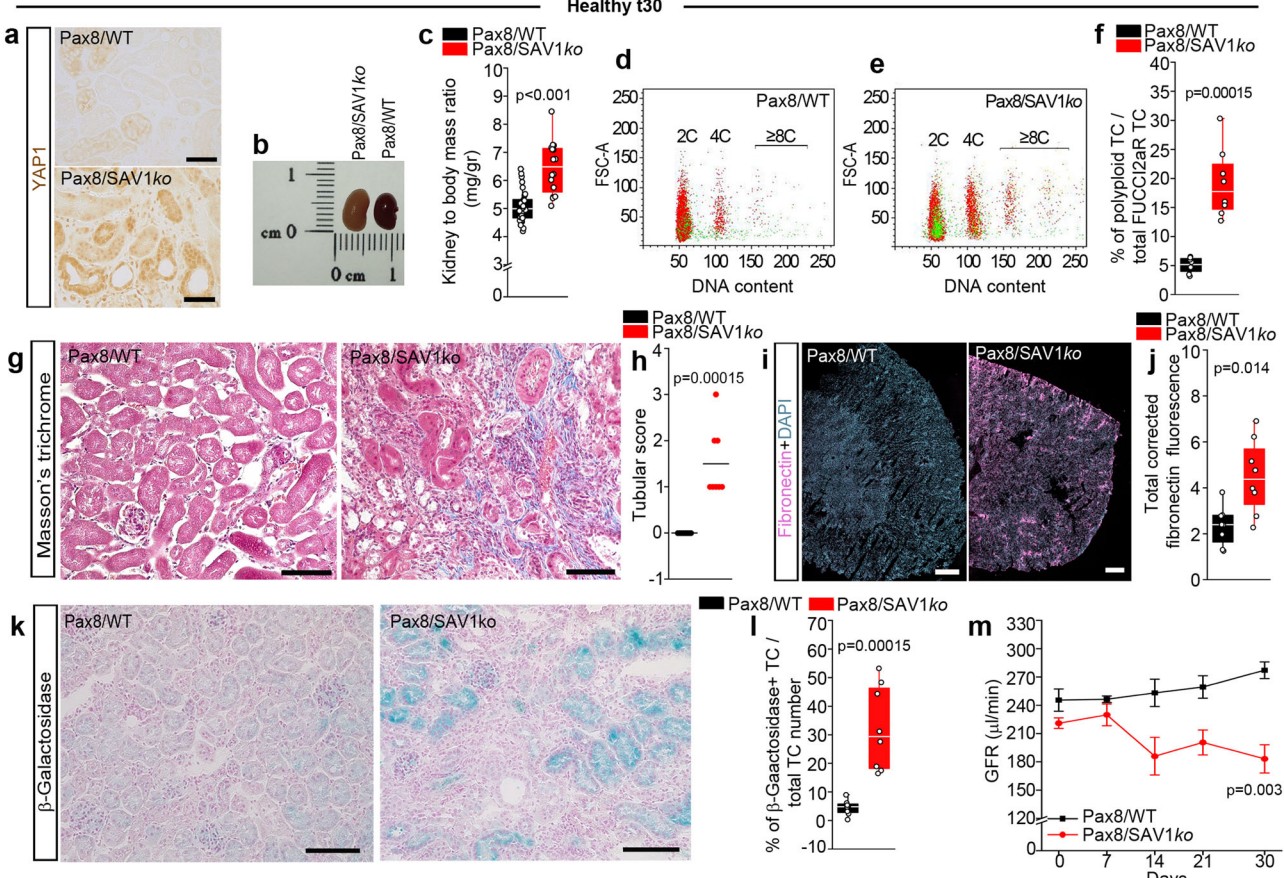

**Fig. 6 | Pax8/SAV1*ko* healthy mice show increased tubular cell (TC) polyploidization and spontaneously progress toward CKD. a** Representative pictures for immunohistochemistry of active-YAP1 staining. Bars 100 μm. A representative experiment out of 4 is shown. **b** Picture of Pax8/FUCCI2aR/SAV1*ko* (Pax8/SAV1*ko*) and Pax8/WT kidneys. **c** Kidney weight in Pax8/WT ($n = 30$) and Pax8/SAV1*ko* mice ($n = 16$), $p = 7.2 \times 10^{-6}$. FACS plots of TC in **d** Pax8/WT and in **e** Pax8/SAV1*ko* mice, showing diploid (2C), tetraploid (4C) and octaploid or greater (≥8C) TC. Colours match the FUCCI2aR reporter ($n = 8$). **f** Percentage of polyploid TC ($n = 8$). **g** Representative pictures of Masson's trichrome staining ($n = 8$). Bars 100 μm. **h** Tubular score evaluated on Masson's trichrome staining ($n = 8$). **i** Sequential scanning of kidney section stained for fibronectin. DAPI counterstains nuclei. Bars 500 μm. **j** Quantification of fibronectin deposition by digital morphometry ($n = 8$). **k** Senescence-associated β-galactosidase assay ($n = 8$). Bars 100 μm. **l** Percentage of β-galactosidase⁺ TC ($n = 8$). **m** GFR measurement for 30 days ($n = 5$). Statistical significance was calculated by two-sided Mann-Whitney test; numbers on graphs represent exact $p$ values or are provided in the legend. Two-way ANOVA test of significance followed by Bonferroni post-test was employed for graph m. Data are expressed as mean ± SEM in graph m. Box-and-whisker plots: line = median, box = 25–75%, whiskers = outlier (coef. 1.5).

polyploid TC (Fig. 9c, d). The total number of FUCCI2aR TC did not differ between the two groups, proving that polyploid TC do not proliferate (Fig. 9e). In addition, mice treated with the YAP1 inhibitor CA3 displayed a better-preserved kidney tissue (Fig. 9f) as well as a significant reduction of senescent TC and fibronectin staining compared to control mice (Fig. 9g–j). We further validated these results in the ischemic injury model (Supplementary Fig. 10a–g).

To exclude any systemic effect of the CA3 administration, we repeated the same experimental design in Pax8/YAP1*ko* mice in which we triggered recombination by doxycycline administration 4 days after nephrotoxic as well as ischemic injury (Fig. 9k–o and Supplementary Fig. 10h–p). Pax8/WT mice recombined at day 4 were used as controls. Delayed recombination of YAP1 prevented CKD progression (Fig. 9k) and reduced further endoreplication cycles of polyploid TC (Fig. 9l, m). Accordingly, mice displayed less fibrosis and senescent TC compared to control mice (Fig. 9n, o and Supplementary Fig. 10h, i). Animal studies have shown that senescence occurs upon AKI, drives CKD and senolytic treatment can avoid AKI to CKD transition[36,37]. By administering senolytic therapy (i.e., dasatinib in combination with quercetin)[36,38] at day 4 after nephrotoxic AKI we observed a significant decrease in the percentage of cycling polyploid TC, in association with

a significantly preserved kidney function at day 30, decreased percentage of senescent TC and fibrosis development (Fig. 9p–t and Supplementary Fig. 10q, r). These results indicate that cycling polyploid TC were responsible for the senescent profibrotic phenotype and were the primary trigger of CKD progression. Thus, YAP1-inhibition initiated right after the early injury phase of AKI, can attenuate ongoing polyploidization of TC, which is sufficient to prevent the long-term trade-off of this survival mechanism, i.e., AKI-CKD transition.

## Discussion

AKI can be induced by dehydration, toxins, drug exposure, trauma or infections, representing one of the ancestral challenges to individual and species survival since the early phases of evolution. Nowadays, dialysis can save patients' life in the acute phase. However, without artificial replacement of kidney function, AKI can be lethal in few hours because of liquid and electrolyte misbalance. Investigating the temporal and spatial distribution and biological significance of TC polyploidy in response to AKI revealed that: 1. in the early phase after AKI, TC polyploidization outside the site of injury is an adaptive stress response to TC loss at the site of injury in order to assure survival by sustaining residual kidney function; 2. during the late phase of AKI,

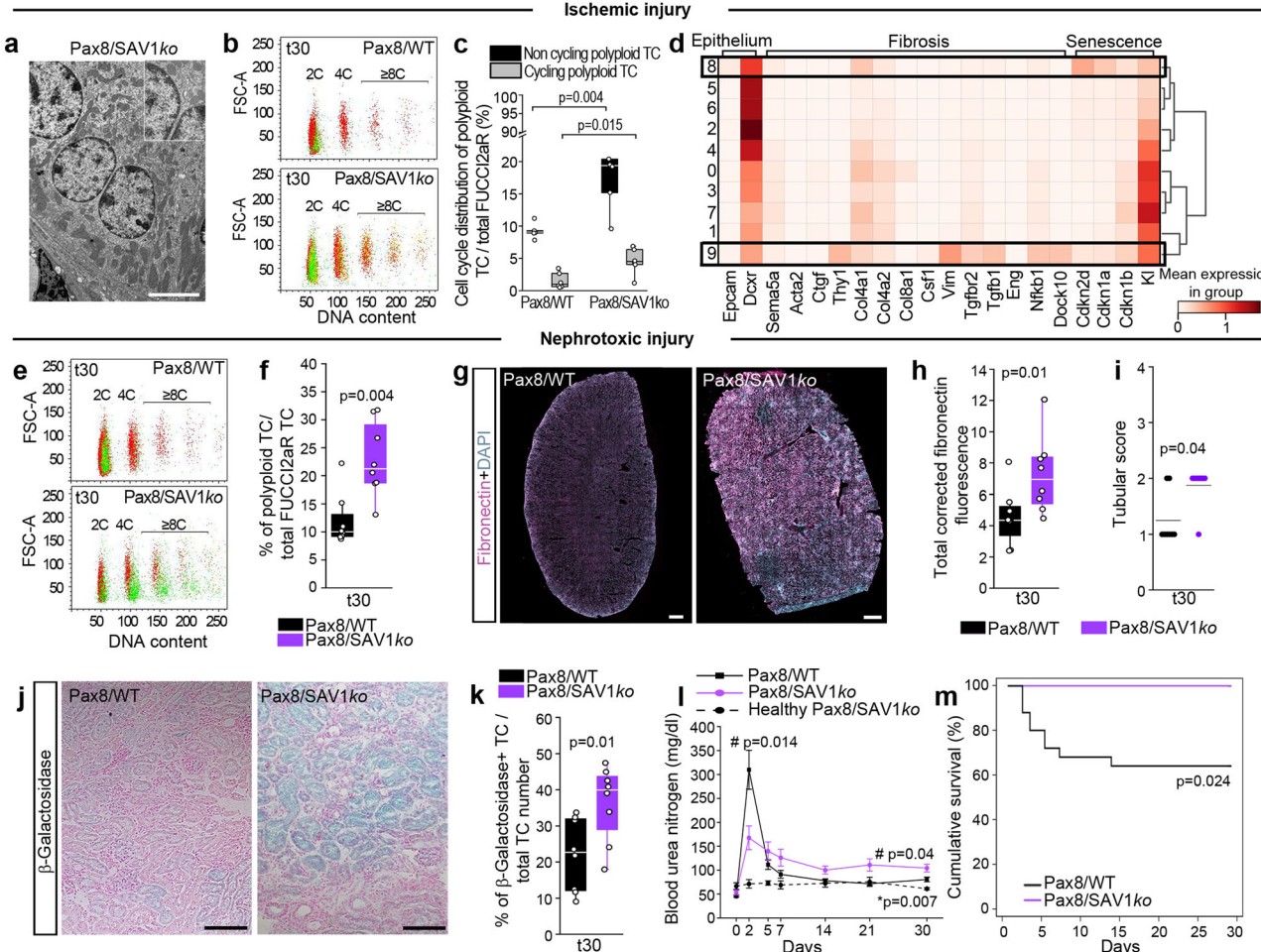

**Fig. 7 | YAP1-driven tubular cell (TC) polyploidization attenuates AKI but aggravates AKI-CKD transition. a** Transmission electron microscopy of a binucleated TC in Pax8/SAV1*ko* mice at day 30 after IRI. Bar 5 μm. **b** FACS plots of TC in Pax8/WT (*n* = 6) and in Pax8/SAV1*ko* mice (*n* = 6) at day 30 after IRI, showing diploid (2C), tetraploid (4C) and octaploid or greater (≥8C) TC. Colours match the FUCCI2aR reporter. **c** Cell cycle distribution of polyploid TC at day 30 after IRI (*n* = 6). **d** Matrix plot showing epithelial, fibrotic and senescence genes in mouse proximal tubular cells at day 2 and 30 after IRI. **e** FACS plots of TC at day 30 (t30) after nephrotoxic injury in Pax8/WT (*n* = 8) and in Pax8/SAV1*ko* mice (*n* = 8). **f** Percentage of polyploid TC after nephrotoxic injury (*n* = 8). **g** Sequential scanning of whole kidney sections stained for fibronectin at day 30 after nephrotoxic injury. DAPI counterstains nuclei. Bars 500 μm. **h** Quantification of fibronectin deposition by digital morphometry after nephrotoxic injury (*n* = 8). **i** Tubular score evaluated on Masson's trichrome staining (*n* = 8). **j** Senescence-associated β-galactosidase assay at day 30 after nephrotoxic injury (*n* = 8). Bars 100 μm. **k** Percentage of β-galactosidase⁺ TC after nephrotoxic injury (*n* = 8). **l** Blood urea nitrogen measurement after nephrotoxic injury (*n* = 10 nephrotoxic Pax8/WT and Pax8/SAV1*ko*, *n* = 6 healthy Pax8/SAV1*ko*). #statistical significance calculated between nephrotoxic Pax8/WT and Pax8/SAV1*ko*, *statistical significance calculated between nephrotoxic and healthy Pax8/SAV1*ko*. **m** Survival analysis after nephrotoxic injury. Kaplan-Meier analysis showed a significant difference at Log rank comparison X2 = 5.07, *p* = 0.024 (*n* = 24 Pax8/WT, *n* = 11 Pax8/SAV1*ko*, none censored). Statistical significance was calculated by two-sided Mann-Whitney test; numbers on graphs represent exact *p* values or are provided in the legend. Data are expressed as mean ± SEM in graph l. Box-and-whisker plots: line = median, box = 25–75%, whiskers = outlier (coef. 1.5).

cycling polyploid TC become senescent and develop a profibrotic phenotype promoting CKD development; 3. blocking polyploidization by YAP1 inhibition in the right window of opportunity avoids CKD after AKI without interfering with the protective activity of TC polyploidy in early AKI; 4. senolytic therapy prevents CKD after AKI by blocking continuous TC polyploidization. Collectively, these results have three important implications:

First, they revise the existing concept on how the kidney responds to AKI which is currently based on staining with "proliferation markers", such as Ki67, PCNA, BrdU that cannot distinguish endo-eplicating from proliferating cells[6–8]. Widespread TC proliferation was considered to mediate tissue reconstitution, driving functional organ recovery after AKI. By contrast, our data dissociate the recovery of kidney function from structural repair at the site of kidney injury. We demonstrate that polyploidization-mediated hypertrophy of TC distant from the injury site provides a life support in the early phase of AKI.

Second, they solve the apparent inconsistency of why even while complete functional recovery upon AKI is frequent, AKI survivors often end up in CKD[1]. The current paradigm refers this phenomenon to a process defined as 'maladaptive repair'[39]. In contrast, our data demonstrate that interstitial fibrosis and CKD represent trade-offs of an acute response required to prevent death in the early AKI phase. Other life-preserving acute stress responses such as the neurohumoral response, the renin-angiotensin-system or mineralocorticoid receptor signaling, also come with the long-term trade-off of promoting chronic organ degeneration and failure[40,41]. Consistently, the role of TC polyploidization in preventing AKI fatality ultimately promotes AKI-CKD transition via TC senescence and interstitial fibrosis.

Third, our study identifies TC polyploidization as a target for therapeutic intervention to attenuate the transition of AKI to CKD with all its consequences, such as progression of CKD and possibly the related development of cardiovascular disease[2–5]. The paucity of valid therapeutic interventions keeps resulting in high mortality rates

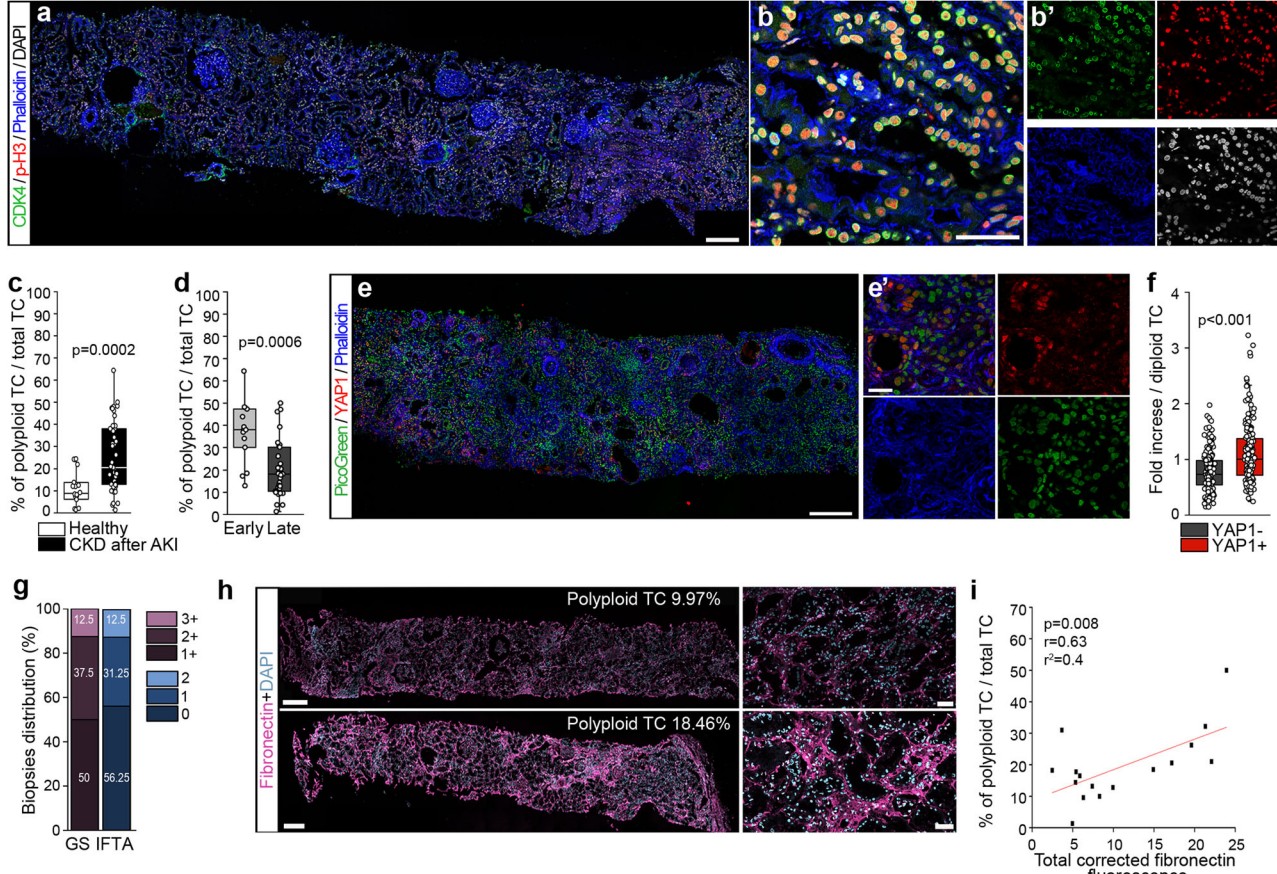

**Fig. 8 | YAP1-related tubular cell (TC) polyploidization correlates with fibrosis in human kidney biopsies. a** Representative sequential scanning of kidney biopsy from a CKD after AKI patient showing CDK4, p-H3 and Phalloidin staining (*n* = 45). DAPI counterstains nuclei. Bar 250 μm. **b**, **b'** Higher magnification of kidney section with split images. Bar 50 μm. **c** Percentage of polyploid TC in CKD after AKI (*n* = 45) *vs* healthy patients (*n* = 18). **d** Percentage of polyploid TC in "early" (*n* = 14) and "late" groups (*n* = 31). **e** Representative sequential scanning of kidney biopsy from a CKD after AKI patient showing YAP1 and Phalloidin staining (*n* = 8). PicoGreen counterstains nuclei. Bar 250 μm. **e'** Higher magnification of kidney section with split images. Bar 25 μm. **f** DNA content quantification of YAP1⁺ nuclei *vs* YAP1⁻ nuclei of TC over diploid TC (*n* = 8, for each biopsy 25 YAP1⁺/Phalloidin⁺ and 25 YAP1⁻/Phalloidin⁺ nuclei respectively were quantified), *p* = 2.19 × 10¹¹. **g** Scoring of glomerular sclerosis (GS) and interstitial fibrosis and tubular atrophy (IFTA) in kidney biopsies of "late" group (*n* = 16). **h** Representative sequential scanning of biopsies stained for fibronectin (*n* = 16). DAPI counterstains nuclei. Bars 250 μm. **i** Linear correlation between polyploid TC and fibronectin deposition quantified by digital morphometry in the "late" group (*n* = 16) was assessed by Pearson correlation coefficient. Statistical significance was calculated by two-sided Mann-Whitney test; numbers on graphs represent exact p values or are provided in the legend. Box-and-whisker plots: line = median, box = 25–75%, whiskers = outlier (coef. 1.5).

during the acute phase and a poor long-term prognosis[1,2]. Current treatment of AKI is still confined to supportive strategies such as dialysis. Promising trials of agents, including diuretics, dopamine and atrial natriuretic peptide, were effective in animal experiments but failed when translated into clinical settings[42]. The results of our study provide an explanation for these observations, showing that the same polyploid TC population that is responsible for hypertrophy and recovery of kidney function immediately after AKI is also responsible for senescence, interstitial fibrosis and CKD in the long run. Indeed, we proved that YAP1 controls TC polyploidization offering a druggable target to block CKD development. However, systemic inhibition of YAP1 does not exclude that cells other than TC could contribute to the observed effect. Nevertheless, the comparable results obtained after triggering YAP1 inactivation selectively in TC after AKI demonstrates that the drug-mediated YAP1-inhibition prevents CKD after AKI by blocking TC polyploidization and subsequent TC senescence and interstitial fibrosis. Interestingly, Zheng et al. reported that YAP1 inhibition is associated with in vivo amelioration of kidney fibrosis and suggested that YAP1 activation after AKI is not compatible with a G2/M cell cycle arrest[43]. Consistently, Gerhardt et al. observed no sign of G2/M cell cycle arrest after AKI using scRNA-seq[44]. Intriguingly, the results of our study suggest that the previously reported cells in G2/M[45,46]

rather represent polyploid cells in G1. Indeed, the method usually applied of measuring DNA content alone cannot distinguish polyploid cells in G1 from diploid cells in G2/M without a simultaneous assessment of the cell cycle phase[45,46]. Nevertheless, even the sophisticated methods applied in this study have a sensitivity limitation and still underestimate the percentage of polyploid TC after AKI for the following reasons: 1. The Confetti reporter misses the polyploid TC that recombine twice or multiple times the same colour; 2. The FUCCI2aR reporter can detect as polyploid only those TC that have completed the cell cycle and are in G1 with a 4C DNA content in a certain moment or as cycling polyploid cells only if their DNA content is >of 4C. Taken together, these results show that polyploidy after AKI is a diffuse phenomenon, explaining its crucial role in the recovery of kidney function in the early phase, as well as in promoting senescence and fibrosis in the late phase after AKI. This implies that treatments for AKI must consider the right window of opportunity because targeting polyploid TC too early would be detrimental for patient survival. Conversely, polyploid TC may become a right target for treatment once the early phase of AKI is bypassed but profibrotic polyploid TC have not yet induced CKD development. We propose that blocking polyploidization at least can protect from CKD development after AKI. Whether other mechanisms in addition to inhibition of TC

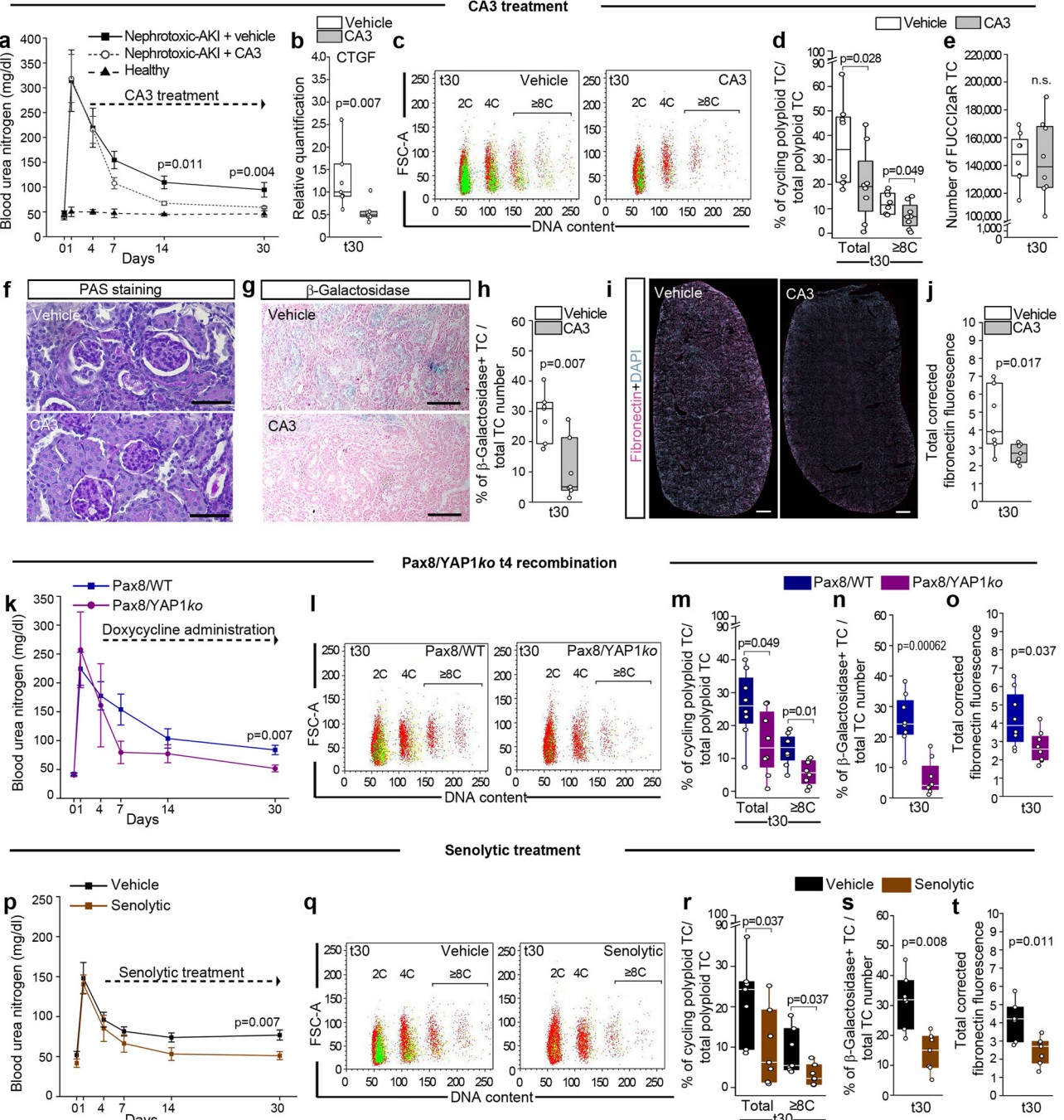

**Fig. 9 | Time-dependent inhibition of tubular cells (TC) polyploidization attenuates AKI-CKD transition. a** Blood urea nitrogen measurement in healthy WT mice (n = 5) and after nephrotoxic injury (n = 7). **b** qRT-PCR for YAP1-downstream target CTGF in nephrotoxic injury after CA3 treatment (n = 7). **c** FACS plots of TC in Pax8/FUCCI2aR mice after nephrotoxic injury, showing diploid (2C), tetraploid (4C) and octaploid or greater (≥8C) TC (n = 8). Colours match the FUC-CI2aR reporter. **d** Percentage of total and ≥8C cycling polyploid TC in Pax8/FUCCI2aR mice after nephrotoxic injury (n = 8). **e** Total number of FUCCI2aR TC after nephrotoxic injury (n = 8). **f** PAS staining in WT mice at day 30 after nephrotoxic injury (n = 7). Bars 50 μm. **g** Senescence-associated β-galactosidase assay in WT mice at day 30 after nephrotoxic injury (n = 7). Bars 100 μm. **h** Percentage of β-galactosidase⁺ TC (n = 7). **i** Sequential scanning of kidney sections stained for fibronectin in WT mice at day 30 after nephrotoxic injury (n = 7). Bars 500 μm. DAPI counterstains nuclei. **j** Quantification of fibronectin deposition by digital morphometry (n = 7). **k** Blood urea nitrogen measurement in Pax8/WT and Pax8/YAP1ko mice after nephrotoxic injury (n = 8). **l** FACS plots of TC after

nephrotoxic injury showing diploid (2C), tetraploid (4C) and octaploid or greater (≥8C) TC (n = 8). Colours match the FUCCI2aR reporter. **m** Percentage of total and ≥8C cycling polyploid TC (n = 8). **n** Percentage of β-galactosidase⁺ TC (n = 8). **o** Quantification of fibronectin deposition by digital morphometry (n = 8). **p** Blood urea nitrogen measurement in Pax8/WT mice after nephrotoxic injury (n = 7). **q** FACS plots of TC in Pax8/FUCCI2aR mice after nephrotoxic injury, showing diploid (2C), tetraploid (4C) and octaploid or greater (≥8C) TC (n = 7). Colours match the FUCCI2aR reporter. **r** Percentage of total and ≥8C cycling polyploid TC (n = 7). **s** Percentage of β-galactosidase⁺ TC (n = 7). **t** Quantification of fibronectin deposition by digital morphometry (n = 7). t30: day 30 after nephrotoxic AKI. t4 recombination: doxycycline administered 4 days after nephrotoxic AKI. Senolytic treatment: quercetin+dasatinib. PAS: Periodic-acid schiff. Statistical significance was calculated by two-sided Mann-Whitney test; numbers on graphs represent exact p values. Data are expressed as mean ± SEM in graph a, k, and p. Box-and-whisker plots: line = median, box = 25–75%, whiskers = outlier (coef. 1.5).

polyploidization contribute to the beneficial effects of YAP1 inhibition after AKI remains to be established and will require further studies.

In summary, our findings revise the current concept of the biological response to AKI, providing a unified model of kidney adaptation to acute tubular necrosis. During injury, TC polyploidization is the fastest way to increase the residual function of differentiated TC. Increasing function without delay is required to avoid AKI fatality during the early phase, that is why this mechanism likely was positively selected during evolution, at the cost of tissue remodeling and chronic dysfunction later in life[47,48]. Nevertheless, YAP1-driven TC polyploidization is an attractive therapeutic target to prevent CKD when blocked during the recovery phase of AKI. This concept may apply also to other organs with a kidney-like low turnover of parenchymal cells.

## Methods

### Mice

To visualize the cell cycle progression of Pax8+ TC, the Pax8.rtTA;TetO.Cre;R26.FUCCI2aR (Pax8/FUCCI2aR) mouse model was employed. This model was obtained by crossing Pax8.rtTA;TetO.Cre mice[8] with mice harboring the Fluorescent Ubiquitin-based Cell cycle Indicator (FUCCI2aR) Cre-dependent reporter (European Mouse Mutant Archive (EMMA), INFRAFRONTIER-I3, Neuherberg-München, Germany), which consists of a bicistronic Cre-activable reporter of two fluorescent proteins whose expression alternates based on cell cycle phase: mCherry-hCdt1 (30/120) (red), expressed by nuclei of cells in G1 phase, and mVenus-hGem (1/110) (green), expressed by nuclei of cells in S/G2/M. Cells can also appear as yellow at the G1/S boundary[49]. Mice were developed on a full C57Bl/6 background[8]. Triple transgenic mice were crossed with either SAV1 knock-out (Sav1$^{tm1.1Dupa}$, JAX:027933) to obtain Pax8/SAV1ko mice or with YAP1 knock-out mice (Yap1$^{tm1.1Dupa}$, JAX:027929) to obtain Pax8/YAP1ko mice both from the Jackson Laboratory. To induce reporter expression, at 5 weeks of age mice were given 4 mg/ml doxycycline hyclate (Merck) in drinking water supplemented with 2.5% sucrose (Merck) for 10 days, followed by a 7-day washout. Un-induced animals showed no leakage nor non-specific transgene expression of the transgenic systems. At the end of the washout period, male mice underwent a unilateral ischemia reperfusion injury (IRI, ischemic injury) of 30 min, and were then sacrificed at day 2, 3, 5 and 30 days post-IRI. Sham operated mice were used as controls. As IRI tolerance is profoundly increased in female mice compared with that observed in male mice, female mice underwent glycerol-induced nephrotoxic AKI (nephrotoxic-AKI, nephrotoxic injury). Mice were sacrificed at day 1, 2 and 30 days after nephrotoxic-AKI. Healthy mice were used as controls. Animals with identical genotype and similar age were assigned to experimental groups in a blinded manner. Successful transgene recombination was assessed at the time of sacrifice by DNA and RNA expression as described below.

For the treatment with YAP1 inhibitor CA3 (CIL56), Pax8/FUCCI2aR mice and non-induced (WT) littermates underwent nephrotoxic-AKI (female mice) or ischemic-AKI (male mice) at 7 weeks of age. Following evaluation of the degree of kidney damage by BUN or GFR measurement, mice were assigned to the treated or control group ensuring evenly distribution in the experimental groups. CA3 (CIL56) was dissolved in DMSO and prior to injection diluted in PBS and sonicated. At day 4 after AKI mice were divided in two groups receiving CA3 (1.5 mg/kg, Selleck, S8661) or vehicle (PBS with 5% DMSO) injection via i.p. every other day until 30 days and then sacrificed[34,35]. To trigger recombination of Pax8/YAP1ko and Pax8/WT mice after ischemic (male) and nephrotoxic (female) AKI, mice at 7 weeks of age were given 4 mg/ml doxycycline hyclate (Merck) in drinking water supplemented with 2.5% sucrose (Merck) at 4 days after AKI for 10 days, followed by a 7-day washout. For the senolytic treatment, Pax8/FUCCI2aR mice underwent nephrotoxic-AKI (female mice) at

7 weeks of age. Following evaluation of the degree of kidney damage by BUN measurement, mice were assigned to the treated or control group ensuring evenly distribution in the experimental groups. At day 4 after AKI mice were divided in two groups receiving Quercetin (50 mg/Kg, Q4951, Merck) + Dasatinib (5 mg/Kg, CDS023389, Merck)[36] or vehicle (PBS with 5% DMSO) via gavage every other day until 30 days and then sacrificed.

The number of mice used, numbers of replicates, and statistical values (where applicable) are provided in the figure legends. Animal experiments were approved by the Institutional Review Board and by the Italian Ministry of Health and performed in accordance with institutional, regional, and state guidelines and in adherence to the National Institutes of Health Guide for the Care and Use of Laboratory Animals. Mice were housed in a specific pathogen-free facility with free access to chow and water and a 12-hour day/night cycle. The references to the ethics approvals are the following: 809/2017-PR; 272/2018-PR; 689/2019-PR; 864/2021-PR and 239/2022-PR.

### Confetti reporter for polyploidy evaluation

The Confetti reporter, upon induction, allowed the stochastic labeling of cells by permanent recombination of a single colour-encoding gene (red, yellow, green, or blue fluorescent proteins, RFP, YFP, GFP, or CFP), with GFP cells occurring at lower frequency than other colours and being extremely rare[8,20]. For this reason, GFP+ TC were not counted and excluded from the calculation. At 7 weeks of age heterozygous male Pax8.rtTA;TetO.Cre;Rosa.Confetti (Pax8/Confetti) mice[8] underwent unilateral ischemia reperfusion injury (IRI) to trigger polyploid TC formation and 13 days after damage they were given 4 mg/ml doxycycline hyclate to activate reporter recombination. After washout, diploid Pax8+ TC (i.e., cells with one set of chromosomes, DNA content 2C) can stochastically express only one of the four colour of the Confetti reporter. In contrast, cells carrying two (DNA content 4C) sets of chromosomes (i.e., polyploid TC) can activate two fluorochromes at the same time, resulting in bi-coloured or mono-coloured TC. Mice were then sacrificed at the end of the 7-day washout. Age-matched control mice received the same induction without undergoing IRI. As GFP+ TC were not included in the analysis, quantitative analysis of polyploid TC was performed by counting TC with the following bi-coloured combinations of fluorescence: RFP + YFP, RFP + CFP, YFP + CFP. The count was normalized on the total number of Confetti TC (GFP+ TC and their combinations were excluded accordingly). Bi-coloured TC were counted by two independent blinded observers. We additionally calculated the efficiency of reporter recombination in our mice, by counting the number of Confetti+ TC over the total TC which is 84.7 ± 1.7%. The percentage of observed bi-coloured TC is 13.3 ± 2.5%. When normalized for the efficiency of reporter recombination, the percentage of bi-coloured TC correspond to 15.7%.

Based on the likelihood of event occurrence, we estimated that 1/3 of polyploid TC with 4C DNA content will recombine twice the same colour and cannot be distinguished from the mono-coloured diploid ones (Supplementary Fig. 2c–g). The acquisition of Pax8/Confetti mice was performed with Leica SP8 STED 3X confocal microscope (Leica Microsystems) with the following set for wavelength excitation: cyan 470 nm, green 488 nm, yellow 514 nm, and red 543 nm. The number of mice used, numbers of replicates, and statistical values (where applicable) are provided in the figure legends.

### Genotyping

Tail biopsies were incubated overnight at 55 °C in lysis reagent (1 M TrisHCl pH 8.5, 0.5 M EDTA, 20% SDS, 4 M NaCl, 0.1 mg/mL proteinase K neutralized with 40 mM TrisHCl, all from Merck), centrifuged and DNA extracted using isopropanol (Merck). Primers and PCR parameters were obtained from Jackson Laboratory online resources of the relative strain purchased or previously reported experimental

procedure[8]. To distinguish transgene homozygosity from heterozygosity, qRT-PCR were performed by using 5 ng/µl of genomic DNA with LightCycler® 480 SYBR Green I Master (Roche Diagnostics). The reactions were performed using a LightCycler® 480 (Roche Diagnostics) with a program consisting of 40 cycles each constituted of an initiation phase at 95 °C for 5 s, annealing phase at 60 °C for 10 s and amplification phase at 95 °C for 10 s, followed by 65 °C for 60 s. The following primers were used:

Pax8.rtTA forward 5′-AACGCACTGTACGCTCTGTC-3′ and reverse 5′-GAATCGGTGGTAGGTGTCTC-3′; TetO.Cre forward 5′-TCGCTGCA TTACCGGTCGATGC-3′ and reverse 5′-CCATGAGTGAACGAACCTGGT CG-3′. TCRα genomic DNA was used as gene housekeeping for relative quantification and was amplified by using the forward 5′-CAAATGTTGCTTGTCTGGTG-3′ and the reverse 5′-GTCAGTCGAGTG-CACAGTTT-3′ primers.

To distinguish FUCCI2aR homozygosity from heterozygosity, PCR was performed with the following primers and parameters: FUCCI-P3 5′-TCCCTCGTGATCTGCAACTCCAGTC-3′, FUCCI-P4 5′-AACCCCAGAT GACTACCTATCCTCC-3′, and FUCCI-P7 5′-GGGGGATTGGGAAGACAAT AGC-3′; 3 min 94 °C, 35 cycles of 94 °C 30 s, 58 °C 30 s and, finally, 72 °C for 30 s. SAV1 and YAP1 genotyping was performed with the following primers and parameters: SAV1 forward 5′-CATAGAAGCAGGGGATT CTGA-3′ and reverse 5′-ACATGCTGACCACAAGCAGA-3′; YAP1 forward 5′- AGG ACA GCC AGG ACT ACA CAG-3′ and reverse 5′- CAC CAG CCT TTA AAT TGA GAA C-3′; 3 min 94 °C, 35 cycles of 94 °C 30 s, 58 °C 30 s and finally, 72 °C for 30 s.

To confirm successful recombination of the alleles (YAP1 and SAV1), a small piece of kidney cortex was excised and RNA and DNA were extracted according to manufacturer′ s instruction (Qiagen). Primers and PCR parameters were obtained from previously published procedures[50,51]. PCR was performed with the following primers and parameters:

SAV1-S1 5′-AGGGGATTCTGACATTTCAGTCAGTT-3′, SAV1-S2 5′-AG TCACATGCTGACCACAAGCAGAA-3′ SAV1-S3; 5′-TGCCATTAAGTGTAA TCACTGG-3′; 3 min 94 °C, 35 cycles of 94 °C 30 s, 56 °C 30 s and finally, 72 °C for 1 min; YAP1-P1 5′-CCATTTGTCCTCATCTCTTACTAAC-3′, YAP1-P2 5′-GATTGGGCACTGTCAATTAATGGGCTT-3′, YAP1-P3: 5′-CAG TCTGTAACAACCAGTCAGGGATAC-3′; 3 min 94 °C, 35 cycles of 94 °C 30 s, 63 °C 30 s and finally, 72 °C for 1 min.

## Quantitative Real-Time PCR

Total RNA was extracted from small kidney cortex pieces at the time of sacrifice in all Pax8/SAV1ko and Pax8/YAP1ko mice to further confirm recombination using the RNeasy Mini Kit (Qiagen). Pax8/WT were used as controls. cDNA was synthesized using High Capacity cDNA Reverse Transcription Kit (Applied Biosystems). Data was normalized on HPRT1 expression. For sorted cells and total mCherry-hPTC, RNA was extracted using RNeasy Microkit (Qiagen) and retrotranscribed using TaqMan Reverse Transcription Reagents (Thermo Fisher Scientific). TaqMan RT-PCR for 18S, CTGF, SEMA5A, CCL2, COLα1, CDKN1A, SERPINE1, YAP1, TAZ, E2F7, E2F8, AKT1, CDK1 and CCNB1 was performed using customized TaqMan assays (Thermo Fisher Scientific) on a 7900HT Fast Real-Time (Applied Biosystem).

## Mouse tissue collection and analysis

All animals were weighted prior to sacrifice by $CO_2$ chamber; kidneys were collected, measured and weighted. Kidney weight was normalized on total body weight. For histological analysis, kidneys were fixed in 10% buffered formalin (Bio-Optica) overnight at room temperature, dehydrated then embedded in paraffin, and 5 µm sections were prepared for Periodic-Acid Schiff staining (PAS, Bio-Optica) and Masson's Trichrome staining (Bio-Optica). The scoring of tissue damage was carried out by two independent blinded observers through a semiquantitative evaluation, based on arbitrary score on Masson's trichrome sections under the optical microscope Leica DM750 (Leica Microsystems, Mannheim, Germany). Whole sections were analysed. The score ranged from 0 to 4+ as follows 0: no changes; 1: damage to <25% of the interstitial area; 2: damage to 25–50% of the interstitial area; 3: damage to 50–75% and 4: damage to >75% of the interstitial area[52]. For confocal analysis of the FUCCI2aR reporter and tubular markers, kidneys were incubated in 4% paraformaldehyde in PBS (both from Merck) for 2 h at 4 °C followed by immersion in a 15% sucrose solution in PBS for 2 h at 4 °C and, subsequently, in a 30% sucrose solution in PBS overnight at 4 °C, then frozen[8]. 10 µm sections of renal tissues were analysed on Leica SP8 STED 3X confocal microscope (Leica Microsystems). For β-galactosidase, fibronectin (1:100, Abcam, ab2413) and Ki67 (1:50, Abcam, ab15580) staining, kidneys were snap-frozen. Only cortical fibrosis was analysed. For β-galactosidase staining, snap-frozen tissues were cut in 10 µm sections and fixed with 2% formalin (Bio-Optica) and 0.2% glutaraldehyde (Merck) in pH6 PBS for 20 min at RT then incubated O/N with X-GAL (Thermo Fisher Scientific, B1690) at a final concentration of 1 mg/ml. The nuclei were counterstained with Nuclear Fast Red (Merck). Sections were then dehydrated and mounted with SafeMount (Bio-Optica) and analysed under the optical microscope Leica DM750 (Leica Microsystems). A total of 5 fields for each conditions were counted. For fibronectin and Ki67 staining, snap-frozen tissues were cut in 10 µm sections and fixed in 4% PFA for 30 min.

## Renal ischemia reperfusion injury

Renal ischemia was performed on male mice as previously described[8,53]. Briefly, mice were anesthetized by intraperitoneal injection of Ketamine (100 mg/kg)/Xylazine (10 mg/kg, Bio98 S.r.L, Milan, Italia). The mouse was placed on a thermostatic station laying on the right side and the body temperature was kept at 37 °C, shaved, and disinfected with Povidone-iodine and an incision of 1–1.5 cm on the skin on the left side was performed. The left kidney was then externalized and the renal artery was clamped for 30 min. After clap removal, the muscle layer was sutured, followed by the closure of the skin wound with metal clips. Immediately after the wound closure, 0.5 ml warm sterile saline (0.9% NaCl) was given subcutaneously to each mouse to rehydrate it. The right contralateral kidney was maintained untouched. Sham-operated mice underwent the same surgical procedure without left renal artery clamping.

## Nephrotoxic AKI

Rhabdomyolysis-induced AKI was performed on female mice, by intramuscular injection with hypertonic glycerol (8 ml/kg body weight of a 50% glycerol solution; Merck) into the inferior hind limbs[8].

## Transcutaneous measurement of glomerular filtration rate

Measurement of the glomerular filtration rate (GFR) was done as previously described[8,53]. Mice where anesthetized with isoflurane and a miniaturized imager device built from two light-emitting diodes, a photodiode, and a battery (Mannheim Pharma and Diagnostics GmbH, Mannheim, Germany) were mounted via a double-sided adhesive tape onto the shaved animals' neck. For the duration of recording (-1.5 h) each animal was conscious and kept in a single cage. FITC-sinistrin (Mannheim Pharma and Diagnostics GmbH,) was resuspended in a sterile saline solution to a final concentration of 30 mg/ml. Injection volume (150 mg/kg) was calculated per each mouse according to the body weight. Prior to the intravenous injection of FITC-sinistrin the skin's background signal was recorded for 5 min. After removing the imager device, the data were analysed using MPD Studio software ver.RC6 (MediBeacon GmbH Cubex41, Mannheim, Germany)[54]. The GFR [µl/min] was calculated from the decrease of fluorescence intensity over time (i.e., plasma half-life of FITC-sinistrin) using a two-compartment model, the animals body weight and an empirical

conversion factor, as previously described[8], using the following formula:

$$GFR\,[\mu l/min] = \frac{14616.8\,[\mu l]}{t_{1/2}(FITC - sinistrin)[min]}\frac{bw\,[g]}{100\,[g]}$$

For each time point, GFR value was normalized on the value at baseline and on the sham value except for healthy Pax8/WT, Pax8/YAP1*ko* and Pax8/SAV1*ko* mice and the measurement at day 2 after AKI in Pax8/WT and Pax8/YAP1*ko*.

### Blood urea nitrogen and Potassium quantification

Kidney function was assessed at different time points by collecting a small amount of blood from mice with a metal lancet from the submandibular plexus in order to measure BUN levels or potassium levels. Blood parameters were measured in EDTA anticoagulated plasma samples using Reflotron (Roche Diagnostics), according to the manufacturer's protocols.

### FACS analysis on kidney tissue

Cell cycle analysis was performed on total FUCCI2aR cells (mCherry[+] and mVenus[+] cells) in Pax8/FUCCI2aR, Pax8/SAV1*ko*, and Pax8/YAP1*ko* mice. Kidneys were processed to obtain a single cell suspension as previously reported[8]. Briefly, kidneys were minced using a scalpel and then incubated at 37 °C for 20 min in 1.5 ml of digestion buffer (300 U/ml Collagenase II and 1 mg/ml Pronase E, Merck). The solution was pipetted up and down with a cut 1000 μl pipette tip every 5 min. The digested kidneys were gently pressed through a graded mesh screen (150 mesh, Merck) and the flow through was washed extensively with HBSS (Thermo Fisher Scientific). Following centrifugation, the pellets was digested again with digestion buffer at 37 °C for 20 min, the suspensions were sheared with a 27-G needle every 10 min. Erythrocytes were lysed with NH4Cl 0.8%. Single-cell suspensions were fixed with 1% PFA for 1 h at RT and with 70% ethanol overnight at 4 °C. Incubation with anti-DsRed (1:25, Clontech, 632496) or isotype control (normal rabbit IgG, 1:250, Thermo Fisher Scientific, 02-6102) was followed by Alexa Fluor 647 goat anti-rabbit (1:100, Thermo Fisher Scientific, A-21245) as secondary antibody to detect mCherry[+] TC, whereas for detection of mVenus[+] TC was used an anti-GFP-488 (1:100, Termo Fisher Scientific, A21311). Cells were then incubated with DAPI (4′,6-diamidino-2-phenylindole, 1:2000, Thermo Fisher Scientific), to perform the DNA content analysis. The assessment of polyploid TC was performed using a MacsQuant instrument (Miltenyi Biotec). Alexa Fluor 647 secondary antibody was excited by a 633 nm laser line, GFP was excited by a 488 nm laser line, DAPI was excited by a laser at 405 nm. Polyploid TC were defined as mCherry[+] or mCherry[+]mVenus[+] cells with a DNA content ≥4C and mVenus[+] with a DNA content ≥8C. Polyploid TC were further divided in non-cycling polyploid TC (mCherry[+] DNA content ≥4C) and cycling polyploid TC (mCherry[+] mVenus[+] cells with a DNA content ≥4C and mVenus[+] with a DNA content ≥8C). Cycling cells were defined as mCherry[+] mVenus[+] with DNA content = 2C and mVenus[+] cells up to 4C DNA content. Dying cells were defined as FUCCI2aR[+] cells with DNA content <2C. To analyse the redistribution of cycling cells at day 2 after AKI, we calculated the redistribution of those cycling cells per each mouse in category (1) cycling, (2) dying, and (3) polyploid as 100%. Cell doublets were excluded from the analysis, as previously reported by us[8] and shown in Supplementary Fig. 1. For urine analysis, multiple urine samples were manually collected from each mouse at t0, t2, and t3 after IRI and pulled together. Staining was performed as described above. Number of FUCCI2aR cells was quantified by using a MacsQuant instrument (Miltenyi Biotec) and normalized on the percentage of induction of each mouse calculated by counting the number of mCherry[+] and mVenus[+] TC on the total TC number on confocal microscope. Data

were analysed by FlowLogic software (FlowLogic 7.2.1, Inivai Technology).

### Immunofluorescence

Confocal microscopy was performed on 10 μm sections of murine renal tissues, on hPTC and 10 μm sections of human kidney biopsies on Leica SP8 STED 3X confocal microscope (Leica Microsystems). For fibronectin acquisition and CDK4/p-H3 staining sequential scanning of whole kidney sections was performed using Leica SP8 STED 3X confocal microscope (Leica Microsystems). Fibronectin deposition was quantified using Image J software (RRID:SCR_003070) and normalized on the section area. The following antibodies were used: anti-aquaporin-1 (AQP1, 1:100, Millipore, AB2219), anti-Tamm-Horsfall (THP, 1:20, Cederlane, CL1032A), anti-aquaporin-2 (AQP2, 1:500, Abcam, Ab62628), anti-GFP polyclonal antibody (1:100, Thermo Fisher Scientific, A-11122), anti-phosphorylated Histone 3 (p-H3, 1:2000, Abcam, ab14955), anti-cyclin dependent kinase 4 (CDK4, 1:50, Santa Cruz Biotechnology, SC-601), Phalloidin-633 and Phalloidin-488 (1:40, Thermo Fisher Scientific, A22284 and A12379), anti-TAZ (1:100, Abcam, ab110239), and anti-active YAP1 (1:500, Abcam, ab205270) staining. Alexa-Fluor secondary antibodies were obtained from Thermo Fisher Scientific. Nuclei were counterstained with DAPI (1:1000, Thermo Fisher Scientific) excited with UV laser. Cell surface area based on Phalloidin-488 staining on mCherry[+] TC of Pax8/FUCCI2aR and Pax8/FUCCI2aR/YAP1*ko* mice were measured using Image J software (RRID:SCR_003070)[8]. The analysis was restricted to PTC.

### Immunohistochemistry

Immunohistochemistry was performed following previously published protocols[53]. Briefly, paraffin-embedded tissues were cut in 5 μm sections, deparaffinized, and rehydrated. The sections were then incubated in 3% H2O2 (Merck) for 10 min to block the endogenous peroxidase activity followed by blocking of endogenous Avidin/Biotin according to manufacturer's instruction (Vector Laboratories, Thermo Fisher Scientific, SP-2001). Antigen retrieval was performed in sodium citrate buffer (10 mM, pH 6) al 95 °C for 40 min followed by 30 min of cooling at RT in deionized water. The sections were then blocked and incubated with the primary antibody anti-active YAP1 (1:500, Abcam, ab205270) or anti-TAZ (1:100, Abcam, ab110239). All the buffers were prepared in PBS + 0.1% Triton X-100 reduced (Merck). 3 3,3′-Diaminobenzidine (DAB, Merck) was used as a peroxidase substrate. Sections were dehydrated and mounted using SafeMount (Bio-Optica). As negative control, primary antibody was substituted with an isotype-matched antibody with irrelevant specificity

### Transmission Electron Microscopy

Kidney tissue samples of about 2 mm³ were obtained from Pax8/WT or Pax8/SAV1*ko* 30 days after IRI, embedded in epoxy resin, and routinely processed for transmission electron microscopy observation. Ultra-thin sections were cut using a LKB-Nova ultramicrotome (LKB, Bromma, Sweden, www.lkb.com), counterstained with uranyl acetate and alkaline bismuth subnitrate, and examined under a JEM 1010 transmission electron microscope (Jeol, Tokyo, Japan, www.jeol.com) at 80 kV.

### Kidney biopsy evaluation in humans

All consecutive patients with different nephropathies who underwent renal biopsy in our Nephrology Units (Nephrology, Dialysis and Transplantation, Azienda Ospedaliero- Universitaria Careggi and Nephrology and Dialysis, Meyer Children's University Hospital, Florence, Italy) between 2012 and 2021 were screened in this study. Exclusion criteria were: (1) Positive immunofluorescence staining suggestive of immune-mediated nephropathy; (2) Less than two creatinine values within 48 h in the period before renal biopsy which does not allow the diagnosis of AKI; (3) Insufficient clinical information

and no or lacking laboratory findings. Following these criteria, 45 renal biopsies from patients (mean age 52 ± 17) that developed CKD after AKI were analysed. 18 healthy kidneys (mean age 59.22 ± 3.65; gender (male) 12/18, 66.6%) were used as controls. Authorization and approval to this study was granted by the Ethical Committee on human experimentation of the Azienda Ospedaliero- Universitaria Careggi (Clinical Trial Center (CTC) AOU Careggi, authorizations: n. OSS_10243) and by the Meyer Children's University Hospital, Florence, Italy (Clinical Trial Office, Meyer, authorizations: n.150/2016). According to the Kidney Disease Improving Global Outcomes (KDIGO) guidelines, patients were classified as CKD if they had preoperative GFR less than 60 mL/min/1.73m2[55]. The estimated glomerular filtration rate (eGFR) was calculated using Chronic Kidney Disease Epidemiology Collaboration equations (CKD-EPI creatinine formula)[56]. CKD after AKI biopsies were divided in "early" and "late" group according to the time elapsed from the AKI episode and the biopsy (2-15 days between the AKI episode and the biopsy, >15 days between the AKI episode and the biopsy). Normal kidney fragments were obtained from the pole opposite to the tumour of patients who underwent radical nephrectomy for localized kidney tumours. Formal consents were obtained by the donors or relatives. The AKI stage was classified according to KDIGO Guidelines[57]. Demographic, clinical and histopathological features of patients who developed CKD after AKI that we analysed in this study are reported in Supplementary Table 3. Polyploid TC were quantified as cells positive for CDK4 and p-H3 (see immunofluorescence section). The scoring of CKD after AKI biopsies was carried out by two independent blinded observers through a semiquantitative evaluation, based on arbitrary score on fibronectin sections. Global glomerulosclerosis (GS) ranged from 1+ to 3+ as follows 1+: GS < 25% of total glomeruli; 2+: GS 25−50% of total glomeruli; and 3+: GS 50−70% of total glomeruli. Cortical interstitial fibrosis and tubular atrophy (IFTA) ranged from 0 to 3 as follows 0: involving <10% of cortical tubulo-interstitial area; 1+: involving 10−25% of cortical tubulo-interstitial area; 2+: involving 26−50% of cortical tubulo-interstitial area; and 3+: involving >50% of cortical tubulo-interstitial area. Fibronectin staining was carried out in biopsies belonging to the "late" group and only where kidney cortical tissue was available.

### Ploidy quantification on kidney biopsies

DNA nuclear content was measured as previously described[58] with some modifications. Biopsies were fixed 20 min in 4% PFA and then incubated with 0.75 μl/mL PicoGreen (Thermo Fisher Scientific) for 15 min at room temperature which was determined to be the best incubation times to obtain a non-saturated signals. Importantly, PicoGreen allows the precise and accurate dsDNA concentration measurements. The sections were then stained as with anti-active YAP1 (1:500, Abcam, ab205270) and Phalloidin-633 (1:40, Thermo Fisher Scientific, A22284) antibody as described in the immunofluorescence section. Biopsies were then imaged using Leica SP8 STED 3X confocal microscope (Leica Microsystems) on sequential scanning of whole kidney biopsies. A healthy biopsy was used as a diploid internal control for ploidy measurements by imaging the biopsy at the same gain and settings. Using Image J software (RRID:SCR_003070) regions were drawn around each nucleus and PicoGreen intensity was measured within each outlined nuclear region. The average background was calculated and subtracted from the measured intensities. The average PicoGreen intensity of diploid healthy tubular cells (2C) were analysed and calculated. The ploidy of YAP1+/Phalloidin+ and YAP1−/Phalloidin+ nuclei was calculated by normalizing the PicoGreen intensity to average value of the diploid cells. For each biopsy 25 YAP1+/Phalloidin+ and YAP1−/Phalloidin+ nuclei respectively were quantified.

### Cell culture, virus transduction, and GapmeRs transfection

Primary hPTC cultures (ATCC-PCS-400-010) were maintained in REGM (Lonza, CC-3190). hPTC were seeded at a density 10^5 cells/6-well. The

following day cells were transduced with a pRetroX-G1-Red (Clontech, 631436) to allow the identification of cells in G1 phase. A MOI of 10 was used (Retro-X™ qRT-PCR Titration Kit, 631453) according to manufacturer's instruction. In this plasmid the cell cycle indicator hCdt1 (30-120) is tagged with the red fluorescent protein mCherry. After transduction cells are referred to as hPTC-mCherry. 48 h after transduction, the media was changed and the cells were split within a week. For confocal imaging, hPTC were seeded in Lab-Tek™ II Chamber Slide™ System (Nunc, Thermo Fisher Scientific) and stained O/N with anti-active YAP1 (1:500, Abcam, ab205270) or anti-TAZ (1:100, Abcam ab110239). Nuclei were counterstained with DAPI (1:1000, Thermo Fisher Scientific).

For FACS analysis, cells were seeded and analysed after 48 h. Cell cycle analysis and gating strategy to exclude cell doublets was performed on total hPTC as previously published with some modifications[8]. Cells were trypsinized (Euroclone) and single-cell suspensions were fixed with 0.25% PFA for 30 min on ice and with 70% ethanol overnight. Incubation with anti-DsRed (1:25, Clontech, 632496) or isotype control was followed by incubation with Alexa Fluor 647 goat anti-rabbit (1:100, Thermo Fisher Scientific, A-21245) as secondary antibody to detect mCherry+ hPTC. In the verteporfin experiment, Alexa Fluor 488 goat anti-rabbit (1:100, Thermo Fisher Scientific, A-11008) was employed. hPTC were then incubated with DAPI (1:2000, Thermo Fisher Scientific) to perform the DNA content analysis and analysed on MacsQuant instrument (Miltenyi Biotec). Polyploid hPTC were defined as mCherry+ cells with a DNA content ≥4C. For verteporfin treatment, a dose response curve was performed and the concentration of 0.6 μM for 48 h was chosen. DMSO (Merck) was used as vehicle control. After 48 h treatment, cells were harvested and analyses by FACS or mRNA was extracted to evaluate YAP1, CTGF, E2F7, E2F8, AKT1, CDK1 and CCNB1 expression. For knock-down experiments, YAP1 (LG00806288-DDA), TAZ (LG00806278-DDA), E2F7 (LG00804017-DDA), E2F8 (LG00804007-DDA) and AKT1 (LG00803998-DDA) antisense LNA-GapmeRs were designed by the Qiagen selection tool (Qiagen, Hilden, Germany) and administered to mCherry-hPTC using Xfect transfection reagent (Takara Bio, Kusatsu, Japan). In detail, 10^5 cells/6-well were seeded and transfected for 6 h with 250 nM antisense LNA-GapmeRs for YAP1, TAZ, E2F7, E2F8 and AKT1 or Negative control (scramble, LG00248643-DFA). The efficacy of mRNA knock-down was evaluated 48 h after transfection by qRT-PCR. Primers are described in the quantitative Real-Time PCR section. After 48 h mCherry-hPTC were harvested and analysed by FACS.

### Chromatin immunoprecipitation assay

Primary hPTC (10^7) treated with DMSO or verteporfin for 48 h were fixed with 1% formaldehyde (Merck) in REGM (Lonza, CC-3190) for 20 min at 4 °C, following by quenching with 0.125 M glycine for 10 min at 4 °C. Further, chromatin was shared into DNA fragment of 500-1000 pb long. 100 μg of chromatin was diluted into ChIP dilution buffer containing protease and phosphatase inhibitor (Merck) and incubated with specific antibodies against YAP1 (10 μg/IP, Novus Biologicals, NB110-58358) or normal rabbit IgG (7 μg/IP Thermo Fisher Scientific, 02-6102) overnight at 4 °C. Antibody-chromatin complexes were recovered with Salmon Sperm DNA/Protein A-Agarose beads (Merck) for 1 h at 4 °C, washed and eluted from the beads with elution buffer. After reverse crosslinking and proteinase K treatment, DNA was extracted and analysed by quantitative qRT-PCR on a 7900HT Fast Real-Time (Applied Biosystem) using specific primers: *CTGF* (F: TGGTGCGAAGAGGATAGGG; R: CGGATTGATCCTGACCCCTTG), *AKT1* (F: GAATGGTTGACTCCCCTCGG; R: GCGGCCAAGAGTGACCTAAA) *E2F7* (F: CCTCCTAATTATTTGCAATTTGCCG; R: GACTGGAAGCCAAAC CAGAA) *E2F8:* (F: ATTCCCCAACTTTGAGCAAGGG; R: GTTGGGGGAA AAGTTCAGCAAC). Enrichment was calculated using the formula: ΔCt= Ct (bound) − [Ct (input) - log2 (Input dilution factor)], where chromatin obtained before immunoprecipitation was used as the input control.

Data are expressed as fold enrichment (FR) of each specific antibody over a negative control antibody relative to negative control; FR = 2 exp - [ΔCt (specific antibody) - ΔCt (normal IgG)].

## Fluorescence-activated cell sorting

hPTC transduced with pRetroX-G1-Red (mCherry-G1) retrovirus were trypsinized (Euroclone) at passage 2 after transduction. The cells were then fixed with PFA 0.25%, 0.5% saponin (Merck) with the addition of 1:25 RNAase inhibitor (Promega, N2615). Then anti-DsRed (1:25, Clontech, 632496) or isotype control was incubated for 1 h at RT followed by 1 h incubation with secondary antibody Alexa Fluor 647 goat anti-rabbit (1:100, Thermo Fisher Scientific, A-21245) to detect mCherry+ hPTC. All the antibodies were diluted in 0,5% saponin (Merck) with the addition of 1:100 RNAase inhibitor (Applied Biosystems, N8080119). All the solutions were diluted in RNAase-free PBS prepared with DEPC water (Merck). The procedure was carried out on ice. Finally, hPTC were incubated with DAPI (1:1000, Thermo Fisher Scientific) to perform the DNA content analysis and sorted on the FACSAria III BD (Bioscience). Alexa Fluor 647 secondary antibody was excited by a 633 nm laser line, DAPI was excited by a 405 nm laser line. Data were analysed by FacsDiva software (Beckman Coulter).

## Single cell RNA-sequencing

hPTC were harvested and filtered using the FlowmiTM Tip strainer (Miltenyi Biotec) to remove clumps and debris, followed by viability assessment with trypan blue (Merck) staining and Propidium Iodide (Miltenyi Biotec)/Calcein AM (Thermo Fisher Scientific) staining. The single cell suspension with at least 98% viability was run on a 10x Chromium Single Cell instrument (10x Genomics) following Manufacturer's instructions, as previously described[53]. 3′ gene expression libraries were constructed as previously described[53] and were sequenced on an Illumina NextSeq550 (Illumina Inc., RRID:SCR_020138).

Wild-type C57Bl/6 mice underwent IRI at 7 weeks of age and were sacrificed 2 (n = 2) and 30 days (n = 3) after AKI. Healthy mouse kidney (n = 1) was used as control. Kidneys were minced into 1-mm pieces with a razor blade and incubated at 37 °C in enzyme dissociation buffer containing 250 U/ml Liberase (Roche) and 40 U/ml DNase I (Merck). After 10 min, the solution was transferred to a Miltenyi C-tube, and the gentleMACS D1 program was run (Miltenyi Biotec). These steps were repeated twice. The reaction was stopped by adding 10% FBS. The solution was then passed through a 40 μm cell strainer. The dissociated cells were then incubated with RBC lysis buffer. We then proceeded to remove the dead cells employing the dead cell removal kit (Miltenyi Biotec, 130-090-101) according to manufacturer's instruction. Cells were then filtered using the FlowmiTM Tip strainer and run on a 10x Chromium Single Cell instrument (10x Genomics). This method generated single cell suspension with greater than 95% viability.

## Bioinformatic analysis

**Preprocessing (hPTC & Mice).** Raw sequencing data were processed using the 10x Genomics Cell Ranger pipeline (version 3.0.1). First, *cellranger mkfastq* demultiplexed libraries based on sample indices and converted the barcode and read data to FASTQ files. Second, *cellranger count* took FASTQ files and performed alignment to the human GRCh38 and mouse mm10 reference genome[59] respectively, and then proceeded with filtering and unique molecular identifier (UMI) counting.

**Quality check and normalization (hPTC & Mice).** Next, by means of scanpy toolbox (v1.7.2)[60] we performed the data analysis starting from quality control on both datasets to remove poor-quality cells and badly detected genes. For hPTC we filtered out cells with a mitochondrial read rate >20% and expressed <350 genes, and for mice, >40% and <500 respectively. Cell-specific biases were normalized by dividing the measured counts by the size factor obtained through the scran *computeSumFactors* method, which implements the deconvolution strategy for scaling normalization[61]. Finally, all counts were log-transformed after addition of a pseudocount of 1. After quality control and filtering, we obtained 11,054 hPTC, 3,625 cells from healthy mouse kidney, 2,690 cells and 2,937 cells from IRI at day 2 and 30 respectively.

**Batch correction, dimensional reduction, and visualization (hPTC & Mice).** Next, we mitigated the batch effect in both datasets through the ComBat method[62] to later proceed with feature selection to keep "informative" genes only used for dimensional reduction through principal component analysis (PCA). The first 50 principle components (PCs) were used to construct a neighborhood graph of observations[63] through the *pp.neighbors* function, which relies on the Uniform Manifold Approximation and Projection (UMAP) algorithm to estimate connectivity of data points.

**Downstream analysis of hPTC.** As first step, we clustered data by *tl.louvain* function at different resolutions (0.5, 1, 1.5, 2) and 0.5 proved to be the best, producing eleven clusters of hPTC in vitro. To annotate the clusters, we obtained and defined the marker genes for each cluster, we ran the *tl.rank_gene_groups* function using the Wilcoxon rank-sum method. Cell cycle analysis was performed by creating two lists of genes associated to the S and G2/M phases based on cell cycle genes previously defined[64]. Next, we performed cell cycle scoring through the *tl.score_cell_cycle_genes* function to score S and G2/M phases. To score a gene list, the algorithm calculates the difference of mean expression of the given list and the mean expression of reference genes. To build the reference, the function randomly chooses a bunch of genes matching the distribution of the expression of the given list. Finally, we used Monocle v. 2.14.0 to generate a trajectory with "selected" clusters to infer the consequentiality of events[65]. For this analysis, we excluded clusters 1 and 6 based on the expression of solute carrier family (Slc) genes, which demonstrated that they were not proximal tubular cells but cells belonging to other portions of nephron. As input to Monocle's Reversed Graph Embedding algorithm, we selected a set of 2188 highly variable genes contributing to the second principle component defined via the PCA, which captures variation in cellular maturation stages.

**Downstream analysis of mice.** At first, we clustered data at different resolutions as for hPTCs and obtained the marker genes for produced clusters. Based on results we kept data at resolution 1, the clusters that either did not express specific cell type genes or expressed marker genes of different cell types have been iteratively subclustered (*louvain* function with resolution equal to 0.5). Next, we proceeded with the analysis by isolating proximal tubule clusters and recalculated the neighborhood graph on the latent space in order to cluster and annotate the data. After clustering, we evaluated the normalized count distribution in order to check the presence of polyploid cells. To this end, we binned the normalized counts and we made a barplot showing the distribution of the 5 obtained bins (2000, 2500, 3000, 3500, 4000) in clusters grouped according to the ribosome distribution. Finally, we used Monocle2, with default parameters and 2939 highly variable genes driving the maturation from t2 to t30, to perform a trajectory analysis and infer the consequentiality of events.

## Statistics and reproducibility

Comparison between groups was performed by two-sided Mann-Whitney test or through the analysis of variance for multiple comparisons (ANOVA for repeated measures) with Bonferroni post hoc analysis or with fisher's exact test (two-tailed p-value). A *p*-value <0.05 was considered statistically significant. Statistical analysis was performed using SPSS (RRID:SCR_002865) and OriginPro (RRID:SCR_

015636) statistical software. Correlation between the percentage of polyploid TC and the fibronectin deposition was tested using Pearson correlation coefficient. Kaplan–Meier estimates were used to generate an overall survival curve for Pax8/WT and Pax8/SAV1ko mice after nephrotoxic injury, and for Pax8/WT and Pax8/YAP1ko mice after nephrotoxic injury. Differences among groups were assessed by log-rank test.

Figures 1e, f, a representative experiment out of 4 is shown. Figures 1n, 7a, a representative experiment out of 2 is shown.

### Reporting summary
Further information on research design is available in the Nature Research Reporting Summary linked to this article.

## Data availability
The authors declare that all data supporting the findings of this study are available within the article and its Supplementary Information Files. Processed data for all human and mouse scRNA-seq libraries generated in this study have been deposited in the Gene Expression Omnibus (GEO) database under accession code GSE212275 for the human dataset and GSE212273 for the mouse dataset. Source data are provided with this paper.

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

## Acknowledgements
This research was funded by the European Union's Marie Sklodowska-Curie fellowship program to L.D.C., grant agreement No. 845774. This study was also funded by the European Research Council (ERC) under the European Union's Horizon 2020 research and innovation program (grant agreement no. 101019891) to P.R and by PRIN: progetti di ricerca di interesse nazionale (2017T95E9X) to P.R. L.D.C. is also the recipient of a L'Oreal-Unesco for women in science award. H.J.A. is supported by the Deutsche Forschungsgemeinschaft (AN372/314-4, 27-1, 30-1).

## Author contributions
L.D.C., E.L., and P.R. designed the study and interpreted the data. L.D.C. performed or supervised all the experiments. E.L. performed the analysis of biopsies and Confetti mice. C.C. and M.D. performed flow cytometry and cell cycle analysis and mouse experiments. R.S. analysed the data from the scRNA-seq analysis. P.D.B. performed the knock-down and ChIP assays and helped with mouse experiments. M.L.A and G.A. designed and performed immunofluorescence and confocal microscopy. B.M. carried out all scRNA-seq and assisted with data analysis. S.L. validated and sequenced the single-cell libraries. A.J.P. and A.M. carried out mouse experiments. L.M. performed cell sorting experiments. M.E.M performed GFR measurement and mouse experiments. M.A., F.G., and F.R. organized patient tissue collection, assisted with statistical analysis, and scored the human samples blinded. G.L.R. performed mouse genotyping and assisted with mouse experiments. D.G. and D.B. performed electron microscopy experiments. L.C. and F.B. assisted with statistical analysis and critically revised the manuscript. A.M. helped with scRNA-seq analysis. F.A. assisted and advised on flow cytometry data interpretation. L.L. and H.J.A. critically revised and edited the manuscript and advised on data interpretation. P.R., E.L., H.J.A., and L.D.C. wrote the manuscript and organized the figures. All authors read and approved the final manuscript.

## Competing interests
The authors declare no competing interests.
