## [Peer Review File · Nature Communications]

Reviewers' Comments:

Reviewer #1:

Remarks to the Author:

In this study, De Chiara and et al. report the role of tubular polyploidization on kidney recovery and repair after two different AKI models. Using different transgenic strategies, and integrating cytometry, confocal microscopy, and single-cell RNA seq (among other techniques) the authors demonstrated that an important proportion of proximal tubular epithelial cells (TEC), specifically S1 and S2 segments enter the cell cycle early after AKI but undergo endoreplication even three days after reperfusion or glycerol administration. Moreover, this polyploidization relies on YAP1 activity, as demonstrated in vitro and in transgenic mice. Interestingly, the authors dissected elegantly the role of YAP1-polyploidization on early recovery of kidney function and on the fibrogenic response, demonstrating that it is an early lifesaving mechanism responsible of cell senescence and tissue fibrosis as a trade-off in the long-term, which might be the first time that early severity of AKI is compellingly dissected from CKD transition. In addition, they studied the presence of endocycling markers on kidney biopsies, indicating a potential relation between this process and human renal fibrosis. This is a sound and well-conducted study, that contributes importantly to the field, however, some issues must be addressed:

Major comments:

1. Introduction: the authors should state very clearly the purpose(s) of this study: Characterize temporal and spatial distribution of TECs undergoing polyploidization during AKI-CKD transition? Evaluate the effect of YAP1-induced polyploidization on early AKI recovery, TEC senescence and tissue fibrosis?. In addition, the references that support the background must be cite the original papers and not revisions.
2. Results, Lines 105-111: The authors indicate that the expected probability of having one-coloured polyploid cells is the same as having multicoloured (three) cells. This is an interesting statement, however, given the reported results, there is little information to make this assumption accurately. First, what was the effectiveness of recombination (% of Pax8+ expressing any colour)? Second, any estimation with this data is very likely incorrect. For example, given three stochastic recombinations in a single cell (three sets of chromosomes), the theoretical probability of observing three different colours is 22.22%, while observing just one colour would be 3.7%. So, if the multicoloured cells correspond to 13.5% of all coloured cells, that would indicate that, approximately 60.75% of all coloured cells have at least three independent recombinations, that is, they are polyploid. According to Figure 1b, this assumption would be false, only 11.5% of cells are polyploid. If the authors still consider that it is possible to estimate the proportion of single-coloured polyploid cells, then please specify the calculations in the supplementary material.
3. It will be nice to study the polyploidization in other segments of the tubular epithelium.
4. Taken together, the authors demonstrate the role of YAP-polyploidization of TEC on early adaptive and late maladaptive post-AKI repair. However, they should recognize that other cells with YAP/TAZ regulation during kidney fibrosis (i.e. macrophages, fibroblasts) could contribute to the observed effect with CA3 treatment, as well as, other cellular mechanisms that are regulated by YAP/TAZ in addition to endoreplication. Considering this recent evidence would enrich their discussion to highlight unanswered questions and propose research directions related to the present findings.
5. The results obtained with the CA3 treatment in the nephrotoxic model are interesting and with clinical potential, however, its effectiveness as a target for therapeutic intervention must be also demonstrated in the IR model.
6. For the human biopsies, is required to show more information about the patients such as: sex, age, baseline renal function, the proportion of patients with pre-existing CKD, the stage of CKD after AKI, etc.
7. Kidney biopsies: Indicate r^2 coefficient to report the size effect.
8. The discussion must be improved. Recent evidence has addressed the function of YAP1 and Hippo pathway in the AKI and AKI to CKD (Zheng, Z., Li, C., Shao, G., Li, J., Xu, K., Zhao, Z., Zhang, Z., Liu, J., Wu, H. 2021. This study, should be discussed in detail.
9. In the discussion section, the following asseveration is not supported for this study: "More severe is the injury, higher is the amount of TC that needs to undergo polyploidization-mediated hypertrophy, and higher is the risk to developing fibrosis, senescence and CKD after AKI. To address this, different periods of ischemia, provoking different degrees of renal injury are required

to probe it.

Minor comments:

1. Introduction: Relevant and more recent evidence of AKI-CKD relationship should be considered in addition to or instead of Lewington et al, 2013 (lines 47-49); i.e.: See EJ, Jayasinghe K, Glassford N, Bailey M, Johnson DW, Polkinghorne KR, Toussaint ND, Bellomo R. Long-term risk of adverse outcomes after acute kidney injury: a systematic review and meta-analysis of cohort studies using consensus definitions of exposure. *Kidney Int.* 2019 Jan;95(1):160-172. Noble RA, Lucas BJ, Selby NM. Long-Term Outcomes in Patients with Acute Kidney Injury. *Clin J Am Soc Nephrol.* 2020 Mar 6;15(3):423-429.
2. Methodology: The following sentence must be appear once and no repeated severa times: "The number of mice used, numbers of replicates, and statistical values (where applicable) are provided in the figure legends".
3. Line 77/Fig 1. It is unclear how the authors determined that 41.8% of cycling TC became polyploid. Is it TC cells with >4C DNA content divided by total mVenus+ at day 3? Is it polyploid TC at day 3 (17%) divided by mVenus+/mCherry+ at day 2 (40.7%)? Similar issue with cycling TC that died (48.3%). Please clarify.
4. Figure 1c. 40 and 50% are inverted in y-axis.
5. Is the Pax8-Confetti construct present in just one chromosome? This is fundamental, and not necessarily obvious, to validate their results in Figure 1k-m.
6. Supplementary Figure 3h. Indicate if it is FSC-A vs. FSC-H.
7. Lines 598-599. Indicate the usage of Monocle2 for pseudotime construction and provide more details of the algorithm and branching functions (as in lines 572-593).
8. Lines 113, 143-144. Although the authors evaluate verteporfin in vitro, at this point, in vivo, the results indicate a YAP1-RELATED polyploidization, rather than -dependent. This distinction is convenient because it provides fluency and connection to the following experiments, where they demonstrate a YAP1-dependent mechanism.
9. Kidney biopsies: Line 194: It is difficult to relate patient outcomes and polyploidization without considering baseline kidney function/fibrosis vs. post-AKI parameters. Then, as a "heterogenous cross-sectional" evaluation, it is more appropriate to say "TC polyploidization in human biopsies correlates with kidney fibrosis after AKI". In addition, it would be convenient to indicate how many patients (if any) had CKD diagnosis before AKI episode.
10. Discussion, lines 253-255, the references of the previous work are missing.

Reviewer #2:

Remarks to the Author:

Chiara et al. investigated the role of polyploidy in tubular epithelial cells in AKI using reporter mouse in two models of kidney injury (ischemia-reperfusion injury and nephrotoxic injury). They demonstrated that polyploid tubular epithelial cells play a protective role in acute phase but are causative cells leading to CKD transition after AKI resolution. And they also showed the presence of this phenomenon in human kidney samples. The authors discovered novel and interesting findings that are expected to make an impact in the nephrology research field. Single cell analysis was well designed and the results were described clearly. I have several comments to this paper which would improve quality of this interesting paper.

Major comment

1. AQP1 is expressed in proximal tubular cells as well as descending thin limbs of Henle. How do you distinguish S1, S2, and S3 segment of proximal tubule in Figure 1o? And could it be possible to show the pattern of multi-colored tubular cells in Pax8/Confetti mice from outer cortex into outer medulla after AKI to visually characterize spatial distribution of polyploidization occurred at distant site from injury?
2. Since a polyploid cell has several times the amount of DNA relative to a diploid cell, it would be expected that they have more amount of transcriptome as well. However, normalization steps could remove this effect if there is no spike-in RNA. This could be partially solved using scRNAseq

data even without spike-in RNA (Song et al 2020 Genome Biology). Could the authors employ the same approach that were used in the cited article to prove the presence of the polyploidy cells in their model and show if cluster 9 in the PT cell line and cluster 8 in the mouse kidney PT clusters are the polyploidy cells?

3. Is there a reason why healthy proximal tubular cells were not included when proximal tubular cells were sub-clustered in Figure 2i. And the authors could also perform trajectory analysis using mouse kidney proximal tubular cells similar to hPTC.

4. In the Introduction, I suggest to describe the reason why conditional reporter mice targeting Pax8+ cells were used for examining the role of polyploidy in AKI. This would enhance the understanding of the readers who was not familiar in this field.

5. In Result section (page 6, line 76), the description of the following sentence was not matched with Figure 1. According to Figure 1c, "at day 3, 41.8%±7.0 of polyploid TC was cycling" seems to be accurate description. And I am confused what "≥4C and ≥8C DNA content" was intended for. → At day 3, 41.8%±7.0 of cycling TC had acquired ≥4C and ≥8C DNA content, suggesting one or even multiple endoreplication cycles (Fig. 1a-c) while very few cells were still cycling (6.7%±2.0).

6. I found that DNA contents are different for the same 2C, 4C, and >8C between two FACS plots, Pax8/WT and Pax8/YAP1 ko, in Figure 3a. Please explain.

7. There was no description in the Result section for Figure 6m. Please describe if necessary.

8. In the method section, the author defined the polyploidy cells and cycling cells. But I think the definition of cycling and non-cycling polyploidy cells is necessary to clarify the finding.

9. How did you analyze cell cycle state in single-cell RNA-seq in Figure 2c? Please describe in the Method.

Minor comment

1. As the previous paper (Nat Commun 2018, PMID 29632300) from the same group annotated Fucci2aR cells, I would suggest using the term, mVenous+ cells (ex. mCherry+mVenous+ cells) instead of GFR+ cells when refer to cells arising from Fucci2aR cells because this cell could be confused with GFR+ cell from Confetti mouse although it was rare.

2. In Result (Page 6, line 90), Fucci2aR reporter was used to detect cell cycle status, not polyploidy in this paper. But the following description was misleading.

→ The Fucci2aR reporter is a ubiquitin-based system that permits detection of polyploid TC at a specific moment....

3. In Discussion section, the sentence in line 260 and line 264 was almost the same.

4. In page 20, line 384, the definition for score 4 was missed. Likewise, in page 25, line 507, the definition for score 4 was also missed.

5. In page 23, line 461, CKD4 was misspelled.

6. In statistical analysis, please describe for statistical analysis for Kaplan-Meier curve and correlation analysis.

7. In Figure 6g, please provide Pearson correlation "r" value.

8. In Supplementary figure 3h, what is assessed for x and y axis? Similarly, please indicate the y axis in supplementary figure 3i.

9. Please describe what the numbers in far right side of supplementary figure 4 indicate.

10. In supplementary table 3, the measurement unit for eGFR was generally "mL/min/1.73m²" in humans. And describe how the GFR was estimated in the method section. In GFR measurement in mouse by transcutaneous approach, the measurement unit for GFR is "ul/min/100g bw" in original paper (PMID 22696603). Thus, I wonder if the normalization by body weight was considered in mouse.

Reviewer #3:

Remarks to the Author:

In this manuscript De Chiara et al. analyze the spatial and temporal distribution of tubular epithelial cells during acute kidney injury and repair, focusing on the occurrence of polyploidy. They describe YAP1 driven polyploidization as immediate compensatory mechanisms to maintain residual kidney function early during AKI, which, however, results in senescence important for AKI-CKD transition. They suggest YAP1 inhibition after the early phase of AKI as a way to prevent AKI-CKD transition without affecting the initially positive effect of YAP1.

The authors provide a novel concept on the pathogenesis of AKI and notably on AKI-CKD transition. While I really like this story and the huge amount of experimental in vivo data, there are some issues that should be addressed:

1) While the data on YAP1 and YAP1-deficiency clearly indicates the protective role of YAP in the early phase of AKI, the manuscript does not mention the additional Yki-ortholog TAZ/WWTR1, which also acts on TEAD TFs and covers similar programs as YAP. Could the authors provide some evidence for (or exclude) the role of TAZ in this context? Pharmacologically targeting YAP might always result in effects on TAZ. Sav-KO will also result in TAZ activation.

2) The authors convincingly show that YAP1 is a critical factor for cell proliferation/hypertrophy/repair upon damage/AKI. However, the conclusion based on the YAP KO model and Fig 1-3 that the YAP1-driven TC polyploidization is a life-saving mechanism is not supported by the data. Yap is an essential regulator of proliferation. And, so far, polyploidy is an important factor, but the latter is not causatively connected to the renal outcome.

What is the mechanism of how Yap affects polyploidy?

3) How exclusive is the expression of "cre" driven by PAX8 in the kidney? Is cre expressed in other tissues as well (e.g., adrenal gland as reported in 2004)? It would be important to check this to exclude any systemic effect in the YAP1 KO. Is there any cre expression in podocytes? (Fig.4a has positive podocytes at the bottom; some blue gloms in Fig 4k).

4) In the "YAP-active" model, the authors can not exclude any YAP independent effects of inactive Hippo signaling. This should be discussed. A transgenic model expressing active YAP (S127A) would be useful.

5) It would be important to analyze YAP1 and/or target gene expression/localization in polyploidy cells of the early human biopsies (Fig. 6) to link this back to YAP.

Minor points:

1) The manuscript is quite challenging to read. Important details are hidden in the method sections. Please explain in Fig 1 or the legend that t2,t3,t5, t30 refer to days after AKI.

2) For a more general readership, results should be explained more in detail (starting with Fig. 1; e.g., how do you differentiate between S1-S3? Pictures in "i-m" remain unclear and are not well described, ...)

3) Calling the Sav-KO "YAP-active" is difficult. Sav/Mst certainly have other targets besides YAP (e.g. TAZ....), and YAP is regulated both hippo-dependent and hippo-independent. "YAP active" implies the use of transgenic YAP1 S-A mutants. Please use the correct name of the line.

4) I would suggest presenting human data and mouse data of Fig 6 in separate figures.

Reviewer #4:

Remarks to the Author:

This is an interesting and novel work addressing the relationship between acute kidney injury and chronic kidney disease. Some concerns and suggestions are provided below.

Comments Relating to YAP

In Fig. 3, the KO mice survival rate is quite low, while the tubular cell death is similar in the YAP KO and WT mice. Consequently, survival does not appear to be dependent on tubular damage per se. Please comment on this discrepancy.

In Fig. 3, animal death is evident during the acute phase (day 2 or 3), while Fig. 5d only shows tubular score at day 30. Similarly, fig. 5m shows WT mice dead mostly at the early acute phase. The above suggest that at the acute phase polyploidization is protective, but the aggravative role in AKI-CKD transition is not clear. Please comment.

In Figures 4 and 5, Pax8/YAP1 active mice demonstrate significant damage even in the absence of an extraneous insult; this makes it difficult to determine whether the injury in fig. 5 is additive or aggravated.

Fig. 5I shows protection in the acute phase, but the worsened renal function (increased BUN) is probably because of the already damaged kidney in YAP1 active mice, and cannot be attributed to the worsening of progression. Please comment.

Figures 4I and 5L show similar % injury in Pax8/YAP1 WT and PAX8/YAP1 active mice, which again suggests that the chronic damage may not be enhanced for the AKI-CKD transition. The spontaneous progression toward CKD in fig. 4 doesn't necessarily reflect aggravation of the AKI-CKD transition.

In the studies presented in Fig. 6, systemic effect cannot be ruled out. For example, immune regulation by YAP and the contribution of an immune response in AKI kidney damage are both well-established. A more relevant approach would be to use YAP WT and KO mice and induce tubular specific KO by doxycycline after insult. Alternatively, intra-renal injection of the inhibitor by osmotic pump for chronic infusion has been reported in a number of publications.

There is also some concern as to whether mixing toxic and ischemic insults is a valid strategy. The role of YAP1 and polyploidization in the ischemic model remains unknown, while ischemia represents the majority of clinical cases.

Overall, the role of tubular YAP1 in AKI-CKD transition lacks supportive evidence. Protection in the acute phase through YAPI signaling has been previously published by the authors (reference 6).

Comments relating to senescence

The potential involvement of poly ploidy and senescence in the transition between AKI and CKD is intriguing. With regard to senescence, the authors should also provide data on the senescence-associated secretory phenotype, since the secretion of factors such as IL6, IL8 and ILbeta (among others) could play a central role in the contribution of tubular cell senescence to kidney injury.

If polyploidy/senescence are involved in initial protection of the kidneys followed by exacerbation of the transition to chronic disease, the authors should consider the testing of various senolytics that have been identified in models of aging. This approach could provide direct evidence for the potential roles of polyploidy/ senescence in this experimental model system as well as the possibility of modulation of these parameters in the clinic.

Reviewer #1:

In this study, De Chiara and et al. report the role of tubular polyploidization on kidney recovery and repair after two different AKI models. Using different transgenic strategies, and integrating cytometry, confocal microscopy, and single-cell RNA seq (among other techniques) the authors demonstrated that an important proportion of proximal tubular epithelial cells (TEC), specifically S1 and S2 segments enter the cell cycle early after AKI but undergo endoreplication even three days after reperfusion or glycerol administration. Moreover, this polyploidization relies on YAP1 activity, as demonstrated in vitro and in transgenic mice. Interestingly, the authors dissected elegantly the role of YAP1-polyploidization on early recovery of kidney function and on the fibrogenic response, demonstrating that it is an early lifesaving mechanism responsible of cell senescence and tissue fibrosis as a trade-off in the long-term, which might be the first time that early severity of AKI is compellingly dissected from CKD transition. In addition, they studied the presence of endocycling markers on kidney biopsies, indicating a potential relation between this process and human renal fibrosis. This is a sound and well-conducted study, that contributes importantly to the field, however, some issues must be addressed:

We thank the Reviewer for the encouraging and positive comments.

Major comments:

1. Introduction: the authors should state very clearly the purpose(s) of this study: Characterize temporal and spatial distribution of TECs undergoing polyploidization during AKI-CKD transition? Evaluate the effect of YAP1-induced polyploidization on early AKI recovery, TEC senescence and tissue fibrosis?. In addition, the references that support the background must be cite the original papers and not revisions.

We thank the Reviewer for these comments. We have now modified the introduction clearly stating the purpose of this study (lines 62-67). In addition, we removed all the reviews and cited the original papers where relevant.

2. Results, Lines 105-111: The authors indicate that the expected probability of having one-coloured polyploid cells is the same as having multicoloured (three) cells. This is an interesting statement, however, given the reported results, there is little information to make this assumption accurately. First, what was the effectiveness of recombination (% of Pax8+ expressing any colour)?

We thank the Reviewer for the comments. We have now calculated the percentage of Pax8 recombination ($84.7 \pm 1.7\%$) and we have included this information in the Methods section (lines 447-448).

Second, any estimation with this data is very likely incorrect. For example, given three stochastic recombinations in a single cell (three sets of chromosomes), the theoretical probability of observing three different colours is 22.22%, while observing just one colour would be 3.7%. So, if the multicoloured cells correspond to 13.5% of all coloured cells, that would indicate that, approximately 60.75% of all coloured cells have at least three independent recombinations, that is, they are polyploid. According to Figure 1b, this assumption would be

false, only 11.5% of cells are polyploid. If the authors still consider that it is possible to estimate the proportion of single-coloured polyploid cells, then please specify the calculations in the supplementary material.

We thank the reviewer for this insightful comment, and we agree that our explanation was not clear. For this reason, we have now added a scheme that better explains the use of heterozygous Pax8/Confetti mice for identification of polyploid cells (see Supplementary Fig. 2c-g). As shown in the Supplementary Fig. 2, the diploid cells recombine only one fluorochrome appearing as mono-coloured cells, while polyploid cells with 4C DNA content recombine two fluorochromes, appearing as bi-coloured or mono-coloured cells. Employing the likelihood of event occurrence in a cell with 4C DNA content and evaluating only three fluorochromes (CFP, RFP, YFP), we can calculate that 2/3 of polyploid cells will recombine two different colours appearing bi-coloured and 1/3 will recombine the same colour twice, appearing mono-coloured, resulting undistinguishable from diploid cells (see Supplementary Fig. 2c-g). Even if this estimation is possible, we agree with the Reviewer that the exact percentage of total polyploid TC is virtually impossible to establish. Thus, following the reviewer observation, we have decided not to indicate a number based on probability calculation. We now restrict our explanation to the rationale of the methods and highlight that the percentage of polyploid cells after AKI is underestimated by either the Confetti or the FUCCI2aR reporter for the following reasons: 1. The Confetti reporter does not allow to detect the polyploid cells that recombine two or more times the same colour, appearing as mono-coloured cells which are undistinguishable from the diploid ones; 2. The FUCCI2aR reporter is ubiquitin-based and can detect as polyploid only cells that have completed the cell cycle and are in G1 with a 4C DNA content in a single moment or as cycling polyploid cells only if their DNA content is > of 4C. Although both tools are the only ones currently available that can accurately detect the presence of polyploid cells, they still underestimate the phenomenon of polyploidy in the kidney. Given the functional relevance of tubular cell polyploidization after AKI, we believe it is important to make the point that polyploidy after AKI is a diffuse phenomenon. This is now better illustrated in the results and discussion (see lines 113-120 in the results section and lines 356-366 in the discussion section).

3. It will be nice to study the polyploidization in other segments of the tubular epithelium.

We thank the Reviewer for the suggestion. We have now performed an additional analysis included in Fig. 1l-o and Supplementary Fig. 2k-m, showing the distribution of polyploid cells in the other segments of the tubular epithelium of the nephron.

4. Taken together, the authors demonstrate the role of YAP-polyploidization of TEC on early adaptive and late maladaptive post-AKI repair. However, they should recognize that other cells with YAP/TAZ regulation during kidney fibrosis (i.e. macrophages, fibroblasts) could contribute to the observed effect with CA3 treatment, as well as, other cellular mechanisms that are regulated by YAP/TAZ in addition to endoreplication. Considering this recent evidence would enrich their discussion to highlight unanswered questions and propose research directions related to the present findings.

We indeed agree that other cells may be influenced by the CA3 treatment. However, the use of Pax8 reporter mice permits to assess the effect exclusively on tubular cells. We have now included additional experiments in which we triggered recombination of Pax8/YAP1^{ko} mice 4 days after ischemic and nephrotoxic AKI. Indeed, these experiments replicated the results obtained with the CA3 treatment, demonstrating that the effect of CA3 specifically involves tubular epithelial cells and excludes significant contributions from other cells. These experiments are now included in Fig. 9k-o and in Supplementary Fig. 10h-p. In addition, as suggested by the Reviewer we have now improved the discussion, highlighting unanswered questions and proposing research directions related to the present findings (lines 345-358, 370-372 and lines 378-379 of the discussion section).

5. The results obtained with the CA3 treatment in the nephrotoxic model are interesting and with clinical potential, however, its effectiveness as a target for therapeutic intervention must be also demonstrated in the IR model.

Following the suggestion of the Reviewer, we performed the requested experiments. Consistent with our other findings, post-ischemic mice that received CA3 starting from 4 days after AKI showed a significant better recovery of kidney function compared to vehicle-treated controls. This was accompanied by a lower percentage of senescent TC and less interstitial fibrosis and, most importantly, a lower percentage of cycling polyploid TC. These results validate the observations made with the nephrotoxic kidney injury model. The data are now included in the revised version of the manuscript as Supplementary Fig. 10a-g.

6. For the human biopsies, is required to show more information about the patients such as: sex, age, baseline renal function, the proportion of patients with pre-existing CKD, the stage of CKD after AKI, etc.

We thank the Reviewer for this insightful comment. We have now included the requested information in Supplementary Table 3.

7. Kidney biopsies: Indicate r/r^2 coefficient to report the size effect.

We have now included the requested information in Fig. 8i.

8. The discussion must be improved. Recent evidence has addressed the function of YAP1 and Hippo pathway in the AKI and AKI to CKD (Zheng, Z., Li, C., Shao, G., Li, J., Xu, K., Zhao, Z., Zhang, Z., Liu, J., Wu, H. 2021. This study, should be discussed in detail.

We have now improved the discussion and included the suggested paper (lines 345-353, ref. 43).

9. In the discussion section, the following asseveration is not supported for this study: "More severe is the injury, higher is the amount of TC that needs to undergo polyploidization-mediated hypertrophy, and higher is the risk to developing fibrosis, senescence and CKD after AKI. To

address this, different periods of ischemia, provoking different degrees of renal injury are required to probe it.

We thank the Reviewer for this insightful comment. We have now removed the sentence.

Minor comments:

1. Introduction: Relevant and more recent evidence of AKI-CKD relationship should be considered in addition to or instead of Lewington et al, 2013 (lines 47-49); i.e.: See EJ, Jayasinghe K, Glassford N, Bailey M, Johnson DW, Polkinghorne KR, Toussaint ND, Bellomo R. Long-term risk of adverse outcomes after acute kidney injury: a systematic review and meta-analysis of cohort studies using consensus definitions of exposure. *Kidney Int.* 2019 Jan;95(1):160-172. Noble RA, Lucas BJ, Selby NM. Long-Term Outcomes in Patients with Acute Kidney Injury. *Clin J Am Soc Nephrol.* 2020 Mar 6;15(3):423-429.

Added as suggested.

2. Methodology: The following sentence must be appear once and no repeated severa times: "The number of mice used, numbers of replicates, and statistical values (where applicable) are provided in the figure legends".

Removed as suggested.

3. Line 77/Fig 1. It is unclear how the authors determined that 41.8% of cycling TC became polyploid. Is it TC cells with >4C DNA content divided by total mVenus+ at day 3? Is it polyploid TC at day 3 (17%) divided by mVenus+/mCherry+ at day 2 (40.7%)? Similar issue with cycling TC that died (48.3%). Please clarify.

We thank the Reviewer for pointing this out. The percentage of cycling cells goes from $39.8 \pm 12.5\%$ at day 2 to $8.8 \pm 4.7\%$ at day 3. This means that the cells that enter the S phase of the cells cycle at day 2 either 1) divided 2) died or 3) became polyploid. Considering this redistribution as 100%, at day 3 $44.4 \pm 15.8\%$ had died, $21 \pm 10.3\%$ had divided (mVenus+ TC with 4C DNA content) and $38.8 \pm 9.2\%$ had become polyploid (mCherry+ TC and mCherry+/mVenus+ TC with $\geq 4C$ DNA content and mVenus+ TC with $\geq 8C$ DNA content). Now we explain this better in the results section (lines 86-105).

4. Figure 1c. 40 and 50% are inverted in y-axis.

Corrected accordingly.

5. Is the Pax8-Confetti construct present in just one chromosome? This is fundamental, and not necessarily obvious, to validate their results in Figure 1k-m.

We thank the Reviewer for this insightful comment. We used only heterozygous Pax8/Confetti mice, i.e. mice carrying the Confetti reporter only on one set of chromosomes, as previously

reported only in the Methods section. To make it clearer for the reader, we have now better specified it also in the results (lines 112-120) and in the methods sections (lines 435-453). We have now added a specification also in the Figure legends (Fig. 1 and Supplementary Fig. 2), and a more complete scheme is now present in Supplementary Fig. 2.

6. Supplementary Figure 3h. Indicate if it is FSC-A vs. FSC-H.

Corrected accordingly.

7. Lines 598-599. Indicate the usage of Monocle2 for pseudotime construction and provide more details of the algorithm and branching functions (as in lines 572-593).

Corrected accordingly (lines 765-768 and 779-781).

8. Lines 113, 143-144. Although the authors evaluate verteporfin in vitro, at this point, in vivo, the results indicate a YAP1-RELATED polyploidization, rather than -dependent. This distinction is convenient because it provides fluency and connection to the following experiments, where they demonstrate a YAP1-dependent mechanism.

Corrected accordingly.

9. Kidney biopsies: Line 194: It is difficult to relate patient outcomes and polyploidization without considering baseline kidney function/fibrosis vs. post-AKI parameters. Then, as a “heterogenous cross-sectional” evaluation, it is more appropriate to say “TC polyploidization in human biopsies correlates with kidney fibrosis after AKI”. In addition, it would be convenient to indicate how many patients (if any) had CKD diagnosis before AKI episode.

We agree with the Reviewer. We have modified the title of the paragraph as suggested and the number of patients with CKD diagnosis before AKI episode is now added in Supplementary Table 3.

10. Discussion, lines 253-255, the references of the previous work are missing. Added as suggested new references 40-42.

Reviewer #2 (Remarks to the Author):

Chiara et al. investigated the role of polyploidy in tubular epithelial cells in AKI using reporter mouse in two models of kidney injury (ischemia-reperfusion injury and nephrotoxic injury). They demonstrated that polyploid tubular epithelial cells play a protective role in acute phase but are causative cells leading to CKD transition after AKI resolution. And they also showed the presence of this phenomenon in human kidney samples. The authors discovered novel and interesting findings that are expected to make an impact in the nephrology research field. Single

cell analysis was well designed and the results were described clearly. I have several comments to this paper which would improve quality of this interesting paper.

We thank the Reviewer for the encouraging comments.

Major comment

1. AQP1 is expressed in proximal tubular cells as well as descending thin limbs of Henle. How do you distinguish S1, S2, and S3 segment of proximal tubule in Figure 1o? And could it be possible to show the pattern of multi-colored tubular cells in Pax8/Confetti mice from outer cortex into outer medulla after AKI to visually characterize spatial distribution of polyploidization occurred at distant site from injury?

We thank the Reviewer for this important specification. We distinguished S1, S2 and S3 segments based on their morphology as well as their localization in distinct areas of the kidney cortex, as previously reported (PMID: 23897681, ref. 21). We now specified this in the results section (lines 131-132).

In addition, we included a new analysis to establish: 1. The spatial distribution of bi-colored tubular cells in Pax8/Confetti mice showing that distribution of polyploid TC mainly occurs distantly from the site of injury (new graph Supplementary Fig. 2j and Fig. 1o); 2. The spatial distribution of bi-coloured tubular cells in other segments of the tubule, i.e. in the thick ascending limb of the Henle's loop and in the distal straight tubule (identified by staining for THP), and in the collecting ducts (identified by staining for AQP2). These data are now included in new Fig. 1o and Supplementary Fig. 2k-m).

2. Since a polyploid cell has several times the amount of DNA relative to a diploid cell, it would be expected that they have more amount of transcriptome as well. However, normalization steps could remove this effect if there is no spike-in RNA. This could be partially solved using scRNAseq data even without spike-in RNA (Song et al 2020 Genome Biology). Could the authors employ the same approach that were used in the cited article to prove the presence of the polyploidy cells in their model and show if cluster 9 in the PT cell line and cluster 8 in the mouse kidney PT clusters are the polyploidy cells?

We thank the Reviewer for this important suggestion. In our study, we employed the 10X Genomics Chromium System technology. This technology allows to isolate thousands of cells but with a very variable capture efficiency (30 to ~80%,) and, uses a non-full-length scRNA-seq approach (3' enrichment), contrary to the authors of the suggested paper. In fact, Song et al. normalized the data through the Transcript per Million (TPM) normalization, a commonly used normalization method for full-length scRNA-seq data (Li et al., 2009, PMID: 20022975) and not for 3' enrichment methods. Collectively, these factors prevent us from employing the same approach used in the cited article. Nevertheless, we adopted a different strategy based on normalized counts binning as shown in new Fig. 3j, k. Importantly, the obtained results proved that ribosomal synthesis was accompanied by a progressive increase of transcriptome abundance along the trajectory and culminating in endoreplicating clusters 8 and 9 in vivo, confirming that polyploid TC are hypertrophic cells.

3. Is there a reason why healthy proximal tubular cells were not included when proximal tubular cells were sub-clustered in Figure 2i.

When we tried to include healthy proximal tubular cells, transcriptome profiles of cells shared between healthy and injured tubular cells directed subclusterization and masked the many functional states that appear only after injury. Excluding healthy cells from the analysis led to a much better visualization of all the functional states induced by injury.

In addition, note that the distribution of clusters in UMAP is now different from the previous version of the manuscript. This is due to an update of the software, which however, does not alter the number and/or composition of the clusters.

And the authors could also perform trajectory analysis using mouse kidney proximal tubular cells similar to hPTC.

We thank the Reviewer for this insightful suggestion. We have now performed the requested trajectory and included it in the current version of the manuscript (new Fig. 3h, i). The trajectory analysis along pseudotime confirmed that tubular cell polyploidization started with increased RNA synthesis and ribosome biogenesis, followed by entry and progression through the cell cycle and culminating with the expression of the endoreplication-specific regulators E2f1, E2f7 and E2f8 and YAP1 target genes.

4. In the Introduction, I suggest to describe the reason why conditional reporter mice targeting Pax8+ cells were used for examining the role of polyploidy in AKI. This would enhance the understanding of the readers who was not familiar in this field.

We thank the Reviewer for this comment. We have now included this explanation into the introduction. Lines 67-69.

5. In Result section (page 6, line 76), the description of the following sentence was not matched with Figure 1. According to Figure 1c, “at day 3, 41.8%±7.0 of polyploid TC was cycling” seems to be accurate description. And I am confused what “≥4C and ≥8C DNA content” was intended for.

→ At day 3, 41.8%±7.0 of cycling TC had acquired ≥4C and ≥8C DNA content, suggesting one or even multiple endoreplication cycles (Fig. 1a-c) while very few cells were still cycling (6.7%±2.0).

We thank the Reviewer for pointing this out. The percentage of cycling cells ranged from 39.8±12.5% at day 2 to 8.8±4.7% at day 3. This implies, that the cells that enter the S phase of the cells cycle at day 2 either 1) divided 2) died or 3) became polyploid. Considering this redistribution as 100%, at day 3, 44.4±15.8% had died, 21±10.3% had divided (mVenus+ TC with 4C DNA content) and 38.8±9.2% had become polyploid (mCherry+ TC and mCherry+/mVenus+ TC with ≥4C DNA content and mVenus+ TC with ≥8C DNA content). Now we explain this better in the results section (lines 86-105).

6. I found that DNA contents are different for the same 2C, 4C, and >8C between two FACS plots, Pax8/WT and Pax8/YAP1 ko, in Figure 3a. Please explain.

Mild differences in intensity of DNA labelling occur between different experiments related to the minimal differences in DNA labelling by the dye. Representative images had now been replaced with images taken all from the same experiment, so that the efficiency of DNA labelling is identical across the analyzed samples.

7. There was no description in the Result section for Figure 6m. Please describe if necessary.

A description of Fig. 6 m (now Fig. 9f) was added, lines 285-288.

8. In the method section, the author defined the polyploidy cells and cycling cells. But I think the definition of cycling and non-cycling polyploidy cells is necessary to clarify the finding.

We corrected the methods accordingly, lines 553-555.

9. How did you analyze cell cycle state in single-cell RNA-seq in Figure 2c? Please describe in the Method.

We added the description in the Methods (lines 756-761). As previously described in Tirosh et al. 2015 (PMID: 27124452, ref.63), we created two lists of genes associated to the S and G2M phases based on cell cycle genes defined and performed cell cycle scoring through the *tl.score_cell_cycle_genes* function.

Minor comment

1. As the previous paper (Nat Commun 2018, PMID 29632300) from the same group annotated FUCCI2aR cells, I would suggest using the term, mVenus+ cells (ex. mCherry+mVenus+ cells) instead of GFR+ cells when refer to cells arising from FUCCI2aR cells because this cell could be confused with GFR+ cell from Confetti mouse although it was rare.

Corrected accordingly.

2. In Result (Page 6, line 90), FUCCI2aR reporter was used to detect cell cycle status, not polyploidy in this paper. But the following description was misleading.

→ The FUCCI2aR reporter is a ubiquitin-based system that permits detection of polyploid TC at a specific moment....

Corrected as requested.

3. In Discussion section, the sentence in line 260 and line 264 was almost the same.

Merged as follows: *“The paucity of valid therapeutic interventions keeps resulting in high mortality rates during the acute phase and a poor long-term prognosis^{1,2}.”* Lines 338-340.

4. In page 20, line 384, the definition for score 4 was missed. Likewise, in page 25, line 507, the definition for score 4 was also missed.

Corrected accordingly (lines 491 and 619).

5. In page 23, line 461, CKD4 was misspelled.

Corrected accordingly (line 570).

6. In statistical analysis, please describe for statistical analysis for Kaplan-Meier curve and correlation analysis.

We have now included the description in the Methods section, "*Statistical analysis*" paragraph lines (790-792).

7. In Figure 6g, please provide Pearson correlation "r" value.

We now included the correlation value, Fig. 8i.

8. In Supplementary figure 3h, what is assessed for x and y axis? Similarly, please indicate the y axis in supplementary figure 3i.

Corrected accordingly

9. Please describe what the numbers in far right side of supplementary figure 4 indicate.

Corrected accordingly.

10. In supplementary table 3, the measurement unit for eGFR was generally "mL/min/1.73m²" in humans. And describe how the GFR was estimated in the method section.

We have now modified the Supplementary Table 3 as requested and included the description in the method section. Lines 604-608.

In GFR measurement in mouse by transcutaneous approach, the measurement unit for GFR is "ul/min/100g bw" in original paper (PMID 22696603). Thus, I wonder if the normalization by body weight was considered in mouse.

Indeed, we had normalized the GFR by body weight. This is now specified in the Methods section (lines 530-532).

Reviewer #3 (Remarks to the Author):

In this manuscript De Chiara et al. analyze the spatial and temporal distribution of tubular epithelial cells during acute kidney injury and repair, focusing on the occurrence of polyploidy. They describe YAP1 driven polyploidization as immediate compensatory mechanisms to maintain residual kidney function early during AKI, which, however, results in senescence important for AKI-CKD transition. They suggest YAP1 inhibition after the early phase of AKI as a way to prevent AKI-CKD transition without affecting the initially positive effect of YAP1.

The authors provide a novel concept on the pathogenesis of AKI and notably on AKI-CKD transition.

We thank the Reviewer for the encouraging comments.

While I really like this story and the huge amount of experimental in vivo data, there are some issues that should be addressed: 1) While the data on YAP1 and YAP1-deficiency clearly indicates the protective role of YAP in the early phase of AKI, the manuscript does not mention the additional Yki-ortholog TAZ/WWTR1, which also acts on TEAD TFs and covers similar programs as YAP. Could the authors provide some evidence for (or exclude) the role of TAZ in this context? Pharmacologically targeting YAP might always result in effects on TAZ. Sav-KO will also result in TAZ activation.

We thank the Reviewer for this insightful comment, which prompted us to perform additional investigation to clarify the role of TAZ. In this revised version of the manuscript, we now report on the role of TAZ modulation in polyploidy response of tubular cells. Specifically:

- 1) We observed no TAZ mRNA accumulation along the pseudotime as demonstrated by scRNAseq. These data are included now in Supplementary Fig. 3g.
- 2) Specifically silencing TAZ mRNA in mCherry-hPTC has no effect on polyploidization, see Supplementary Fig. 3m, n.
- 3) Immunofluorescence on verteporfin treated mCherry-hPTC shows limited effect on TAZ nuclear localization, see Supplementary Fig. 3l
- 4) Immunohistochemistry analysis and quantification on kidneys of Pax8/WT mice at different days after AKI shows no TAZ upregulation, see Supplementary Fig. 5c, d.
- 5) Immunohistochemistry analysis and quantification on kidney tissues of Pax8/YAP1 ko and Pax8/SAV1 ko mice shows no TAZ upregulation, see Supplementary Fig. 6i-p and 7g, h.

Collectively, these results exclude a role for TAZ and confirm the role of YAP1 in modulating polyploidy of tubular cells after AKI.

2) The authors convincingly show that YAP1 is a critical factor for cell proliferation/hypertrophy/repair upon damage/AKI. However, the conclusion based on the YAP KO model and Fig 1-3 that the YAP1-driven TC polyploidization is a life-saving mechanism is not supported by the data. Yap is an essential regulator of proliferation. And, so far, polyploidy is an important factor, but the latter is not causatively connected to the renal outcome.

We thank the Reviewer for this specification. However, we can certainly exclude an effect of YAP1 on proliferation of tubular cells at day 1 and 2 after AKI. Indeed, the number of tubular cells is not significantly different between Pax8/YAP1 ko mice and Pax8/WT mice at day 1 and 2 after AKI as shown in Fig. 5i. Conversely, the percentage of polyploid cells and the size of the kidney are dramatically different (Fig. 5m-o). This would indicate that the impaired kidney functionality observed in Pax8/YAP1 ko mice as well as the reduced mouse survival can only

be related to a defect in polyploidization mediated hypertrophy of tubular cells. To better clarify this point we added some explanations to the Result section (lines 215-218).

What is the mechanism of how Yap affects polyploidy?

We thank the Reviewer for pointing this out, giving us the chance to investigate the mechanism through which YAP1 affects polyploidy of tubular cells. New in vitro experiments using CHIP assay and targeted silencing revealed that YAP1 modulates polyploidy of tubular cells via AKT1, E2F7 and E2F8, all YAP1-specific downstream effectors. These data are now included in a new paragraph (line 182-195) and a new Fig. 4.

3) How exclusive is the expression of “cre” driven by PAX8 in the kidney? Is cre expressed in other tissues as well (e.g., adrenal gland as reported in 2004)? It would be important to check this to exclude any systemic effect in the YAP1 KO.

As already reported by Traykova-Brauch M et al. Nat Med 2012 (PMID 18724376, ref. 17), in adult mice, the Pax8 promoter is exclusively activated in kidney tubular cells. In addition, our transgenic mouse model is conditional, i.e., cre recombination occurs only upon exposure to doxycycline at 5 weeks post-natal and after 10 days doxycycline is withdrawn, which makes it impossible that “cre” recombination occurs in other cells. Thus, we can exclude the expression of the “cre” by other organs and tissues.

Is there any cre expression in podocytes? (Fig.4a has positive podocytes at the bottom; some blue gloms in Fig 4k).

Again, in the kidney only tubular epithelial cells express the Pax8 promoter, thus the “cre” is not expressed by podocytes. What the Reviewer has observed in Fig. 4a (that is now Fig.6a) is endogenous YAP1 expression. Likewise, Fig. 4k (that is now Fig. 6k) shows the presence of β -galactosidase staining, i.e., senescent cells independently from the recombination triggered by the “cre”.

4) In the “YAP-active” model, the authors can not exclude any YAP independent effects of inactive Hippo signaling. This should be discussed. A transgenic model expressing active YAP (S127A) would be useful.

As already mentioned above, we can exclude YAP1 independent effects of inactive Hippo signaling mediated by TAZ in modulating the polyploid response of tubular cells for the following reasons:

- 6) We observed no TAZ mRNA accumulation along the pseudotime as demonstrated by scRNAseq. These data are included now in Supplementary Fig. 3g.
- 7) Specifically silencing TAZ mRNA in mCherry-hPTC has no effect on polyploidization, see Supplementary Fig. 3m, n.
- 8) Immunofluorescence on verteporfin treated mCherry-hPTC shows limited effect on TAZ nuclear localization, see Supplementary Fig. 3l

- 9) Immunohistochemistry analysis and quantification on kidneys of Pax8/WT at different days after AKI mice shows no TAZ upregulation, see Supplementary Fig. 5c, d.
- 10) Immunohistochemistry analysis and quantification on kidney tissues of Pax8/YAP1^{ko} and Pax8/SAV1^{ko} mice shows no TAZ upregulation, see Supplementary Fig. 6i-p and 7g, h.

Collectively, these results exclude a role for YAP1-independent effects of inactive Hippo signaling mediated by TAZ and confirm the role of YAP1 in modulating polyploidy of tubular cells after AKI.

5) It would be important to analyze YAP1 and/or target gene expression/localization in polyploidy cells of the early human biopsies (Fig. 6) to link this back to YAP.

We thank the Reviewer for this suggestion. To address this issue, we performed an analysis of YAP1 positive and YAP1 negative tubular cells in human early biopsies, showing that DNA content is significantly increased in YAP1+ nuclei in comparison to YAP1- nuclei indicating that where YAP1 is active the cells have an increased DNA content (i.e., they are polyploid) (Fig. 8e, f). These data were obtained by staining the section for YAP1 and quantifying DNA content with PicoGreen as previously reported (see PMID: 26958853, ref. 58 and Methods section, lines 627-643) and are now included in Fig. 8e, f.

Minor points:

1) The manuscript is quite challenging to read. Important details are hidden in the method sections. Please explain in Fig 1 or the legend that t2,t3,t5, t30 refer to days after AKI.

Following the suggestion of the Reviewer, we included all the relevant information within the main text and corrected the legend of Fig. 1 and all the other figure legends accordingly.

2) For a more general readership, results should be explained more in detail (starting with Fig. 1; e.g., how do you differentiate between S1-S3? Pictures in “i-m” remain unclear and are not well described, ...)

We thank the Reviewer for the suggestion. We distinguished S1, S2 and S3 segments based on both their morphology and localization in distinct areas of the kidney cortex, as previously reported (PMID: 23897681, ref. 21). We have now specified this in the Results section (Lines 131-132).

3) Calling the Sav-KO “YAP-active” is difficult. Sav/Mst certainly have other targets besides YAP (e.g. TAZ....), and YAP is regulated both hippo-dependent and hippo-independent. “YAP active” implies the use of transgenic YAP1 S-A mutants. Please use the correct name of the line.

Corrected manuscript and figures accordingly.

4) I would suggest presenting human data and mouse data of Fig 6 in separate figures.

We agree with the Reviewer. We have now separated Fig. 6 in Fig. 8 and Fig. 9.

Reviewer #4 (Remarks to the Author):

This is an interesting and novel work addressing the relationship between acute kidney injury and chronic kidney disease. Some concerns and suggestions are provided below.

We thank the Reviewer for the positive comment.

Comments Relating to YAP

In Fig. 3, the KO mice survival rate is quite low, while the tubular cell death is similar in the YAP KO and WT mice. Consequently, survival does not appear to be dependent on tubular damage *per se*. Please comment on this discrepancy.

We thank the Reviewer for pointing this out. Indeed, we believe that survival rate is not dependent on tubular damage *per se* but rather affected by the inability of the surviving tubular epithelial cells to properly compensate the loss of kidney function from injured cells. Obviously, this immediate response to AKI requires YAP1-mediated polyploidization (i.e., hypertrophy) of the surviving tubular cells all across the cortex. Indeed, tubular cell death is similar in Pax8/YAP1*ko* and Pax8/WT mice at day 1 and 2 after AKI (Fig. 5h), but the percentage of polyploid cells and the size of the kidney are dramatically different (Fig. 5m-o). Since Pax8 is selectively activated in tubular cells, the reduced survival of Pax8/YAP1*ko* mice can only be related to a defect in polyploidization mediated hypertrophy of tubular cells. To better clarify this point we added some explanations to the Result section (lines 215-218).

In Fig. 3, animal death is evident during the acute phase (day 2 or 3), while Fig. 5d only shows tubular score at day 30. Similarly, fig. 5m shows WT mice dead mostly at the early acute phase. The above suggest that at the acute phase polyploidization is protective, but the aggravative role in AKI-CKD transition is not clear. Please comment.

In Figures 4 and 5, Pax8/YAP1 active mice demonstrate significant damage even in the absence of an extraneous insult; this makes it difficult to determine whether the injury in fig. 5 is additive or aggravated.

We thank the Reviewer for this insightful comment. We have now directly compared the percentage of polyploid cells, interstitial fibrosis and senescence of tubular cells in the Pax8/SAV1*ko* healthy vs Pax8/SAV1*ko* IRI mice. This comparison shows a significant exacerbation effect of AKI on polyploidy, tubular cell senescence, and interstitial fibrosis. We added these new data to Supplementary Fig. 8 and described them in the results sections (lines 241-244).

Fig. 5I shows protection in the acute phase, but the worsened renal function (increased BUN) is probably because of the already damaged kidney in YAP1 active mice, and cannot be attributed to the worsening of progression. Please comment.

Thanks for pointing this out. In order to dissect the AKI-related aggravation of CKD in Pax8/SAV1*ko* mice, we performed a novel time-course experiment in Pax8/SAV1*ko* mice not undergoing AKI. By directly comparing BUN levels, we show that AKI significantly worsens CKD in Pax8/SAV1*ko* mice. We added this result in Fig. 7I and commented as requested (lines 251-254).

Figures 4I and 5L show similar % injury in Pax8/YAP1 WT and PAX8/YAP1 active mice, which again suggests that the chronic damage may not be enhanced for the AKI-CKD transition. The spontaneous progression toward CKD in fig. 4 doesn't necessarily reflect aggravation of the AKI-CKD transition.

See previous comments. We have now added Supplementary Fig. 8 to address this issue.

In the studies presented in Fig. 6, systemic effect cannot be ruled out. For example, immune regulation by YAP and the contribution of an immune response in AKI kidney damage are both well-established. A more relevant approach would be to use YAP WT and KO mice and induce tubular specific KO by doxycycline after insult. Alternatively, intra-renal injection of the inhibitor by osmotic pump for chronic infusion has been reported in a number of publications.

Thank you very much for the excellent suggestion, which prompted us to perform additional analyses. We indeed agree that CA3 treatment may influence other cells. Following the Reviewer's suggestion, we have now included additional experiments in which we induced Pax8 tubule-specific YAP1*ko* by doxycycline 4 days after ischemic and nephrotoxic AKI (i.e. at the same time when we started CA3 treatment in other experiments). These experiments faithfully replicated the results obtained with CA3 treatment, demonstrating that the effect of CA3 treatment is mediated by tubular cells and not by other cells. These experiments are now included in Fig. 9k-o and in Supplementary Fig. 10h-p.

There is also some concern as to whether mixing toxic and ischemic insults is a valid strategy. The role of YAP1 and polyploidization in the ischemic model remains unknown, while ischemia represents the majority of clinical cases.

We agree with the Reviewer and following his/her suggestion we investigated the role of YAP1 and polyploidization also in the ischemia-reperfusion injury model on: 1. Pax8/WT and wild-type non induced mice treated with the CA3 inhibitor starting at day 4 after ischemic injury. 2. Pax8/YAP1*ko* mice treated with doxycycline to induce recombination of YAP1 at day 4 after ischemic injury. The results obtained in these models confirm those already obtained with the nephrotoxic kidney injury model. This further demonstrates that polyploidization of tubular cells after AKI is the crucial, conserved mechanism of response to acute injury and the driver of AKI-CKD transition. These results are now included in the new Supplementary Fig. 10.

Overall, the role of tubular YAP1 in AKI-CKD transition lacks supportive evidence. Protection in the acute phase through YAPI signaling has been previously published by the authors (reference 6).

We think that the data now added to this revision demonstrate a direct role of YAP1 in polyploidy. Moreover, in the paper quoted by the Reviewer, we described the presence of polyploid cells after AKI but we did not address the mechanisms involved in polyploidization and not even the role of YAP1.

Comments relating to senescence

The potential involvement of poly ploidy and senescence in the transition between AKI and CKD is intriguing. With regard to senescence, the authors should also provide data on the senescence-associated secretory phenotype, since the secretion of factors such as IL6, IL8 and ILbeta (among others) could play a central role in the contribution of tubular cell senescence to kidney injury.

We thank the Reviewer for this suggestion. Data on the SASP of polyploid cells are now added, demonstrating the SASP phenotype of polyploid TC and confirming the senescent state of these cells. These data are now included in Supplementary Fig. 9.

Is polyploidy/senescence are involved in initial protection of the kidneys followed by exacerbation of the transition to chronic disease, the authors should consider the testing of various senolytics that have been identified in models of aging. This approach could provide direct evidence for the potential roles of polyploidy/ senescence in this experimental model system as well as the possibility of modulation of these parameters in the clinic.

We thank the Reviewer for this insightful comment. Following the suggestion, we induced nephrotoxic AKI in Pax8/WT mice and treated them with the senolytic agents quercetin+dasatinib from day 4 after AKI. Consistent with previously published data, we observed that senolytic treatment prevented AKI to CKD transition. Importantly, this beneficial effect was associated not only with senescence reduction, but also with the reduction of polyploidization. These results provide further evidence for the role of polyploidy/senescence in AKI to CKD transition. These data are now included in Fig. 9p-t and Supplementary Fig. 10q, r.

Reviewers' Comments:

Reviewer #1:

Remarks to the Author:

In the revised version of this manuscript, De Chiara and colleagues attended to the previous observations nicely, and the additional experiments support the alleged role of YAP1 and polyploidization on AKI and AKI-CKD. The authors improved the fluency along the manuscript, making it slightly easier for the reader to follow their hypotheses, methodology and results. I only suggest verifying the following issues:

1. It is very unusual that Figures 2e, 4b, Suppls 3m, 3o, 6j, 6n and 7e have all a p value of 0.028. If there is a mistake this will hardly change conclusions, however, please double check that the information along the manuscript is accurate.
2. Is there any reason, methodological or theoretical, to report % of change in GFR in Figure 5d, total GFR in Figure 6m and BUN in most of the rest? Please provide the absolute values of GFR.
3. The GFR values in PAX8WT and Pax8/SAV1ko (Fig. 6m) are too low to the previous values reported (doi:10.1152/ajprenal.00279.2012, doi: 10.14814/phy2.15211), please verify them. In addition, a more detailed description of GFR measurement by FITC-sinistrin must be added, including the volume intravenously injected in each mouse.

Reviewer #2:

Remarks to the Author:

The Authors have addressed all of my concerns with the original manuscript.

Reviewer #3:

Remarks to the Author:

The authors addressed all of my concerns in this revised MS and in their response. Therefore, I fully support publication of this article.

Reviewer #4:

Remarks to the Author:

The authors have conscientiously addressed all of the concerns raised in the previous review.

Reviewer #1 (Remarks to the Author):

In the revised version of this manuscript, De Chiara and colleagues attended to the previous observations nicely, and the additional experiments support the alleged role of YAP1 and polyploidization on AKI and AKI-CKD. The authors improved the fluency along the manuscript, making it slightly easier for the reader to follow their hypotheses, methodology and results.

Thank you very much for your positive evaluation.

I only suggest verifying the following issues:

1. It is very unusual that Figures 2e, 4b, Suppls 3m, 3o, 6j, 6n and 7e have all a p value of 0.028. If there is a mistake this will hardly change conclusions, however, please double check that the information along the manuscript is accurate.

Thanks for pointing this out. The identical p value (0.028) is due to the sample size (n=4) in those graphs and to the stringency of the Mann-Whitney test of significance. We have thoroughly checked the information along the manuscript and corrected all the mistakes which are now tracked in red.

2. Is there any reason, methodological or theoretical, to report % of change in GFR in Figure 5d, total GFR in Figure 6m and BUN in most of the rest? Please provide the absolute values of GFR.

Thanks for pointing this out. We have now provided the absolute values for the GFR in graph 5d. In general, GFR measurement was performed for the unilateral (ischemic) injury while BUN was performed in the case of bilateral (glycerol-induced) injury. In fact, as the ischemic injury is unilateral, BUN does not accurately capture the kidney functionality variations. Likewise, for the Pax8/WT and the Pax8/SAV1ko healthy mice represented in graph 6m, GFR was chosen because the kidney function decrease is subtler and BUN is not able to capture small variations. Taking into consideration the principle of the 3Rs and in an effort to minimize the number of mice employed, BUN was preferred to the GFR measurement unless absolutely necessary.

3. The GFR values in PAX8WT and Pax8/SAV1ko (Fig. 6m) are too low to the previous values reported (doi:10.1152/ajprenal.00279.2012, doi: 10.14814/phy2.15211), please verify them. In addition, a more detailed description of GFR measurement by FITC-sinistrin must be added, including the volume intravenously injected in each mouse.

Thank you for pointing this out. Indeed, the measurements are lower than the previously reported ones because unlike the paper cited by the reviewer, where they show the GFR measurement without body weight normalization ($\mu\text{l}/\text{min}/100\text{g}$) which was calculated as follows:

$$\text{GFR } [\mu\text{l}/\text{min}/100\text{gr}] = \frac{4616.8 \text{ } [\mu\text{l}/100\text{g bw}]}{t_{1/2}(\text{FITC-sinistrin})[\text{min}]}$$

we reported the GFR measurement normalized to the body weight ($\mu\text{l}/\text{min}$) using the following formula

$$\text{GFR } [\mu\text{l}/\text{min}] = \frac{4616.8 \text{ } [\mu\text{l}]}{t_{1/2}(\text{FITC-sinistrin})[\text{min}]} \times \frac{\text{bw [g]}}{100[\text{g}]}$$

We have now included this formula in the material and methods. FITC-sinistrin was resuspended at a final concentration of 30 mg/ml. The volume of FITC-sinistrin injected was then calculated per each

mouse according to the body weight as now indicated in the methods to inject 150 mg/kg of FITC-sinistrin.

Reviewer #2 (Remarks to the Author):

The Authors have addressed all of my concerns with the original manuscript.

Thank you very much for your positive evaluation.

Reviewer #3 (Remarks to the Author):

The authors addressed all of my concerns in this revised MS and in their response. Therefore, I fully support publication of this article.

Thank you very much for your positive evaluation.

Reviewer #4 (Remarks to the Author):

The authors have conscientiously addressed all of the concerns raised in the previous review.

Thank you very much for your positive evaluation.